# *ZC3H11A* mutations cause high myopia by triggering PI3K-AKT and NF-κB-mediated signaling pathway in humans and mice

Chong Chen[1,2,3†], Qian Liu[1,2,3†], Cheng Tang[1,2,3†], Yu Rong[1,2,3], Xinyi Zhao[1,2,3], Dandan Li[1,2,3], Fan Lu[1,2,3]*, Jia Qu[1,2,3]*, Xinting Liu[1,2,3]*

[1]National Engineering Research Center of Ophthalmology and Optometry, Eye Hospital, Wenzhou Medical University, Wenzhou, China; [2]National Clinical Research Center for Ocular Diseases, Eye Hospital, Wenzhou Medical University, Wenzhou, China; [3]State Key Laboratory of Ophthalmology, Optometry and Visual Science, Eye Hospital, Wenzhou Medical University, Wenzhou, China

*For correspondence:
lufan62@eye.ac.cn (FL);
qujia@eye.ac.cn (JQ);
liuxt@eye.ac.cn (XL)

†These authors contributed equally to this work

Competing interest: The authors declare that no competing interests exist.

## eLife Assessment

This work investigates ZC3H11A as a cause of high myopia through the analysis of human data and experiments with genetic knockout of Zc3h11a in mouse, providing a **useful** model of myopia. The evidence supporting the conclusion is still **incomplete** in the revised manuscript as the concerns raised in the previous review were not fully addressed. The article would benefit from a more robust genetic analysis and comprehensive presentation of human phenotypic data to clarify the modes of inheritance in the families, currently limited by loss of patient follow-up and addressing whether there is a reduction in bipolar cell number or decreased marker protein expression through cell counts or quantifiable, less saturated Western blots. The work will be of interest to ophthalmologists and researchers working on myopia

**Abstract** High myopia (HM) is a severe form of refractive error that results in irreversible visual impairment and even blindness. However, the genetic and pathological mechanisms underlying this condition are not yet fully understood. From a cohort of 1015 patients with HM in adolescents, likely pathogenic missense mutations were identified in the *ZC3H11A* gene in four patients by whole exome sequencing. This gene is a zinc finger and stress-induced protein that plays a significant role in regulating nuclear mRNA export. To better understand the function and molecular pathogenesis of myopia in relation to gene mutations, a *Zc3h11a* knockout (KO) mouse model was created. The *Zc3h11a*[+/-] mice exhibited significant shifts in refraction toward myopia. Myopia-related factors, including *Tgfβ1*, *Mmp2*, and *Il6*, were found to be upregulated in the retina or sclera, and electro-retinography and immunofluorescence staining results showed dysfunction and reduced number of bipolar cells in the retina. Transmission electron microscopy findings suggest ultrastructural abnormalities of the retina and sclera. Retinal transcriptome sequencing showed that 769 genes were differentially expressed, and *Zc3h11a* was found to have a negative impact on the PI3K-AKT and NF-κB signaling pathways by quantitative PCR and western blotting. In summary, this study characterized a new candidate pathogenic gene associated with HM and indicated that the ZC3H11A protein may serve as a stress-induced nuclear response trigger, and its abnormality causes disturbances in a series of inflammatory and myopic factors. These findings offer potential therapeutic intervention targets for controlling the development of HM.

## Introduction

Myopia is a highly prevalent eye affliction that commonly develops during childhood and early adolescence. The prevalence of myopia among the adult population ranges from 10% to 30%, while parts of East and Southeast Asia have reported rates as high as 80–90% among young people (*Baird et al., 2020*; *Holden et al., 2016*). High myopia (HM) is a severe refractive error characterized by a diopter ≤−6.00 D or an axial length greater than 26 mm. It is estimated that by 2050, the number of people with HM worldwide will reach 938 million, accounting for 9.8% of the total population. HM can trigger a range of adverse ocular changes, such as cataracts, glaucoma, retinal detachment, macular degeneration, and possibly total blindness (*Koga et al., 2014*; *Saw et al., 2005*). While conventional methods for the prevention and control of myopia can provide some correction, they are not entirely effective in managing its progression, particularly during childhood and adolescence.

With the development of next-generation sequencing, whole exome sequencing (WES) and whole genome sequencing have extended the findings of linkage studies to identify potential causes of syndromic HM (sHM) and non-syndromic HM (nsHM). To date, approximately 20 genes with causal associations have been identified in sHM, including *ZNF644, SCO2, CCDC111, LRPAP1, SLC39A5, LEPREL1, P4HA2, OPN1LW, ARR3, BSG, NDUFAF7, CPSF1, TNFRSF21, DZIP1, XYLT1, CTSH, GRM6, LOXL3,* and *GLRA2* (*Haarman et al., 2022*; *Tian et al., 2023*; *Yang et al., 2023*; *Ye et al., 2023*). These discoveries have provided insight into the molecular mechanisms underlying HM. However, known candidate genes can only explain about 20% of the causes of this disease (*Cai et al., 2019*; *Tedja et al., 2019*). At the same time, the neuromodulators and signal molecules of HM are extremely complex, including sclera extracellular matrix remodeling and endoplasmic reticulum stress (*Ikeda et al., 2022*), inflammatory responses, the release of dopamine and gamma-aminobutyric acid, or abnormalities in myopia-related signaling pathways, such as retinoic acid signaling, TGF-β signaling, and HIF-1α signaling.

Screening mutations from an in-house adolescents HM survey cohort by WES, we identified zinc finger CCCH domain-containing protein 11A (*ZC3H11A*) as an HM candidate gene. This particular gene is a member of the zinc finger protein gene family. Multiple zinc finger protein genes (e.g. *ZNF644, ZC3H11B, ZFP161, ZENK*) are associated with myopia or HM. Of these, *ZC3H11B* (a human homolog of *ZC3H11A*) and five GWAS loci (*Schippert et al., 2007*; *Shi et al., 2011*; *Szczerkowska et al., 2019*; *Tang et al., 2020*; *Wang et al., 2004*) correlate with AL elongation or HM severity. Proteomic studies further suggest ZC3H11A involvement in the TREX complex, implicating RNA export mechanisms in myopia pathogenesis. Additionally, it has been suggested that this protein is involved in stress-induced responses (*Younis et al., 2018*). Dysfunction of *ZC3H11A* results in enhanced NF-κB signaling through defective IκBα protein expression, which is accompanied by upregulation of numerous innate immune- and inflammation-related mRNAs, including *IL8, IL6,* and *TNF* in vitro (*Jimi et al., 2019*). Moreover, patients with myopia have a higher proportion of inflammation-associated cells, including neutrophils, while those with moderate myopia show strained immune system function (*Lin et al., 2016*; *Qi et al., 2022*). Conversely, some traditional Chinese medicines can control myopia progression by suppressing AKT and NF-κB-mediated inflammatory reactions (*Chen et al., 2022*). However, the precise pathological mechanism of *ZC3H11A* in myopia development remains unclear, necessitating further research.

The current study identified four variants in the *ZC3H11A* gene among an HM adolescent cohort of 1015 adolescents. Additionally, *Zc3h11a*[+/-] mice were constructed using the CRISPR/Cas9 system and their myopic phenotypes, visual function, bipolar cell apoptosis, and retinal and scleral microstructure were assessed. Moreover, RNA sequencing and expression experiments were performed to identify perturbed molecules and pathways and to examine the interactions between these factors in relation to HM. The above results will provide new ideas for the prevention and control of HM, especially early-onset, uncorrectable, and familial myopia.

## Results

### ZC3H11A mutations are associated with HM in a Chinese cohort

The genomic data is sourced from an independently established genetic cohort of HM patients. Four missense mutations in the *ZC3H11A* gene (c.412G>A, p.V138I; c.128G>A, p.G43E; c.461C>T, p.P154L; and c.2239T>A, p.S747T) were identified in the 1015 HM patients aged from 15 to 18 years.

The uncorrected visual acuity and axial length of these patients are presented in *Table 1*. All of the identified mutations exhibited very low frequencies in the Genome Aggregation Database (gnomAD) and ClinVar, and using pathogenicity prediction software SIFT, PolyPhen2, and CADD, most of them display high pathogenicity levels (*Table 1*). Among them, c.412G>A, c.128G>A, and c.461C>T were located in or around a domain named zf-CCCH_3 (*Figure 1A and B*). Furthermore, all of the mutation sites were located in highly conserved amino acids across different species (*Figure 1C*). Four mutations resulted in a higher degree of conformational flexibility and altered the negative charge at the corresponding sites (*Figure 1D and E*).

## *Zc3h11a⁺ᐟ⁻* mice exhibited myopic phenotypes

*Zc3h11a⁺ᐟ⁻* mice were constructed against a C57BL/6J background using CRISPR/Cas9 technology (*Figure 2—figure supplement 1*). Retinal fundus images and ocular histomorphology of *Zc3h11a⁺ᐟ⁻* mice at 8 weeks postnatal were assessed against those of their wild-type (WT) counterparts. No significant structural differences were observed (*Figure 2—figure supplement 2*). These findings suggest that the deletion of *Zc3h11a* does not alter the retinal structure or ocular histomorphology. Therefore, *Zc3h11a⁺ᐟ⁻* mice are a relevant model for the investigation of the role of *Zc3h11a* in refractive development.

Refraction and axial length in *Zc3h11a⁺ᐟ⁻* mice were found to be significantly greater than in WT littermates (independent samples t-test, $p<0.05$; *Figure 2A and B*). The difference between the two genotypes was statistically significant at weeks 4 and 6. Correspondingly, the vitreous chamber depth of *Zc3h11a⁺ᐟ⁻* mice was deeper than that of WT littermates (independent samples t-test, $p<0.05$; *Figure 2C*). There were no significant differences in the anterior chamber depth, lens diameter, and body weight between the two groups (*Figure 2D–F*). These results suggest that the myopia phenotype of *Zc3h11a⁺ᐟ⁻* mice is increased.

## Reduced b-wave amplitude and bipolar cell-labeled protein abundance in *Zc3h11a⁺ᐟ⁻* mice

To confirm if *Zc3h11a* is responsible for refractive development regulation, visual function was assessed by electroretinography (ERG). Upon dark adaptation, b-wave amplitudes in 7-week-old *Zc3h11a⁺ᐟ⁻* mice were significantly lower at dark 3.0 (0.48 log cd·s/m²) and dark 10.0 (0.98 log cd·s/m²) compared to WT mice (*Figure 3A and C*). On the contrary, there were no differences in a-wave amplitudes between the *Zc3h11a⁺ᐟ⁻* and WT groups (*Figure 3B*). Based on the ERG results, variations between *Zc3h11a⁺ᐟ⁻* and WT littermates can be observed. Under the dark adaptation conditions of 3.0 and 10.0, there was a significant change in the amplitude of ERG b-waves, indicating impaired bipolar cell function. Immunofluorescence analyses were performed on frozen retinal sections to investigate this phenomenon further. Specifically, Zc3h11a, PKC-α, Opsin-1, and Rhodopsin markers were utilized to detect the Zc3h11a protein, rod-bipolar cells, cone cells, and rod cells, respectively (*Figure 3D–F*). Quantitative analysis of immunofluorescence staining results revealed that the protein abundance of Zc3h11a and PKCα was significantly reduced in the *Zc3h11a⁺ᐟ⁻* group (*Figure 3G and H*). Western blot analysis further showed that the expression level of PKC-α protein in the retina of *Zc3h11a⁺ᐟ⁻* mice was significantly reduced, indicating a decrease in the number of bipolar cells (*Figure 3I and J*). However, there were no significant differences in the protein abundance of Opsin-1 and Rhodopsin between the two groups (*Figure 3—figure supplement 1*). Thus, both the function and number of retinal bipolar cells were decreased in *Zc3h11a⁺ᐟ⁻* mice.

## Retinal ultrastructure alterations in *Zc3h11a⁺ᐟ⁻* mice model

Transmission electron microscopy (TEM) was used to analyze retinal ultrastructure in 10-week-old mice. In the inner nuclear layer (INL) of the retina, in contrast to WT mice, *Zc3h11a⁺ᐟ⁻* mouse cells had enlarged perinuclear gaps (black arrow), perinuclear cytoplasmic edema (blue arrow), and thinned and lightened cytoplasm (*Figure 4A and B*). However, in the outer nuclear layer, no significant difference in cell morphology was observed between the two groups of mice (*Figure 4C and D*). These findings indicate structural damage to bipolar cells in *Zc3h11a⁺ᐟ⁻* mice. Furthermore, when compared to the WT group, the *Zc3h11a⁺ᐟ⁻* mice exhibited relatively damaged photoreceptor cell membrane discs (MB), in which the outer layer was detached and sparsely distributed locally (*Figure 4E and F*). There were also a small number of broken membrane discs, as well as some disorganized and loosely

**Table 1.** The clinical features and mutations of affected patients.

| Patient information | | | Refraction (D) | | Axial length (mm) | | Variation information | | | Prediction software and databases | | | | |
|---|---|---|---|---|---|---|---|---|---|---|---|---|---|---|
| ID | Sex | Age | OD | OS | OD | OS | Genotype | chromosomal positions | Existing_variation (rs numbers) | SIFT (score) | PolyPhen2 (score) | CADD (score) | gnomAD | Clinvar |
| 1 | M | 18 | –6.50 | –5.125 | 25.68 | 25.34 | c.412G>A;p. Val138Ile, Het | chr1:203798692–203798692 | rs142418357 | D (0.03) | PB (0.912) | 24.3 | 7.95E–06 | – |
| 2 | F | 15 | –4.25 | –6.125 | 24.61 | 25.11 | c.128G>A;p. Gly43Glu, Het | chr1:203787771–203787771 | – | D (0) | PS (0.906) | 26 | – | – |
| 3 | M | 17 | –8.00 | –8.75 | 26.16 | 27.02 | c.461C>T;p. Pro154Leu, Het | chr1:203798741–203798741 | – | D (0) | PS (0.628) | 29.3 | – | – |
| 4 | F | 15 | –6.25 | –3.70 | 25.53 | 24.86 | c.2239T>A;p. Ser747Thr, Het | chr1:203821333–203821333 | – | T (0.1) | PS (0.838) | 22.7 | – | – |

M, male; F, female; Het, heterozygote; OD, ocular dexter; OS, oculus sinister; D, deleterious; T, tolerated; PB, probably_damaging; PS, possibly_damaging; gnomAD, Genome Aggregation Database; AF, allele frequency; –, not applicable.

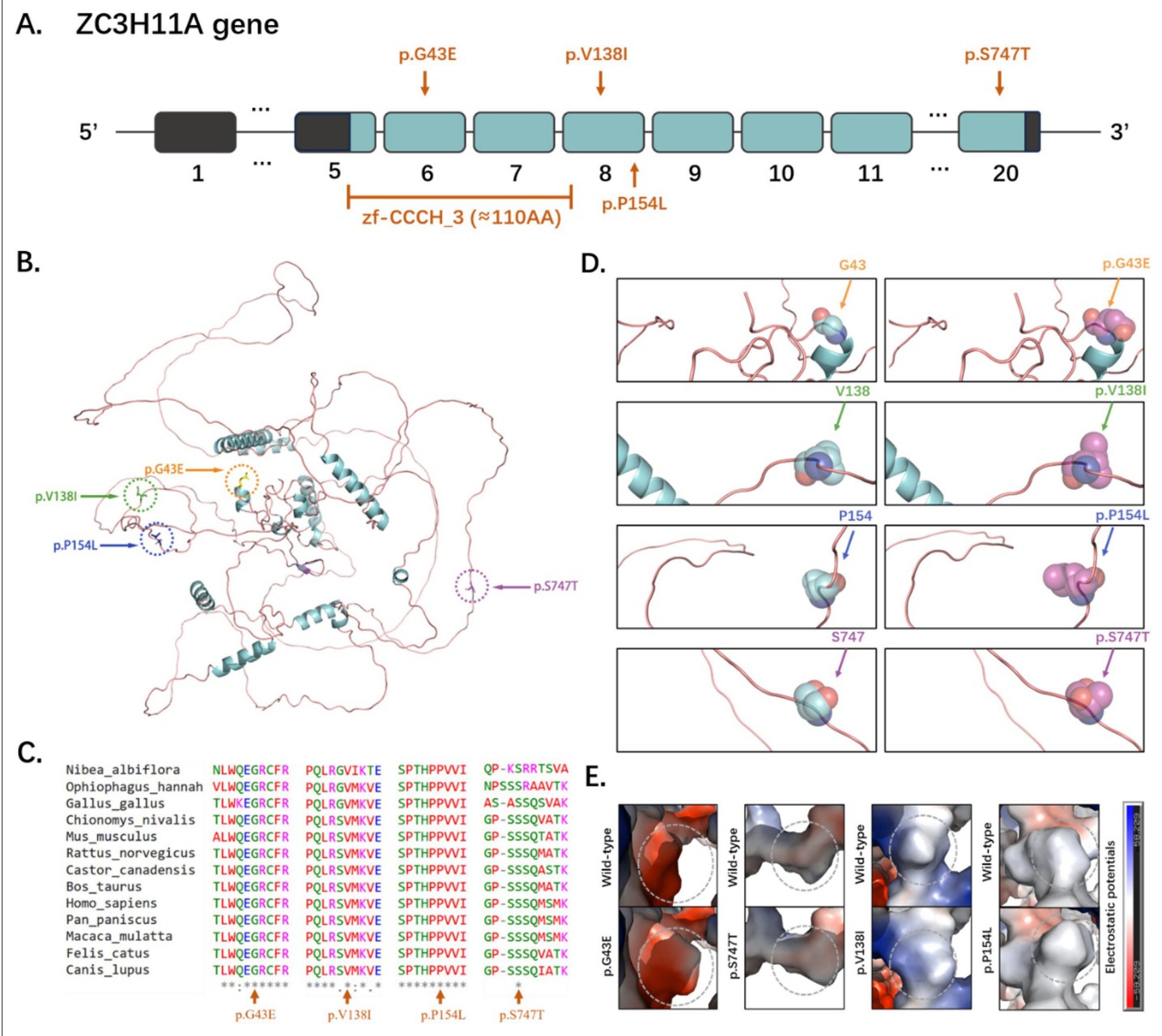

**Figure 1.** Structural and evolutionary analysis of *ZC3H11A* mutations in high myopia. (**A**) Genomic organization of *ZC3H11A*: Exon-intron structure (exons 1–20) with four identified missense mutations (orange arrows: c.128G>A/p.G43E, c.412G>A/p.V138I, c.461C>T/p.P154L, c.2239T>A/p.S747T). Domain architecture showing the zf-CCCH_3 zinc finger domain (exons 5–8) harboring three mutations (G43E, V138I, P154L). (**B**) Full-length ZC3H11A structural model: Predicted tertiary structure (PyMOL v2.5) with mutation sites highlighted: G43E (exon 6, orange), V138I/P154L (exon 8, green and blue), S747T (exon 20, purple). (**C**) Mutation localization: Schematic mapping of mutations to exons: G43E (exon 6), V138I/P154L (exon 8), S747T (exon 20). Exon sizes scaled proportionally. (**D**) Cross-species conservation: Multiple sequence alignment of *ZC3H11A* orthologs showing absolute conservation of mutated residues. (**E**) DynaMut2-predicted conformational flexibility changes: These mutations can result in a higher degree of conformational flexibility at the corresponding sites, potentially destabilizing the structural domain.

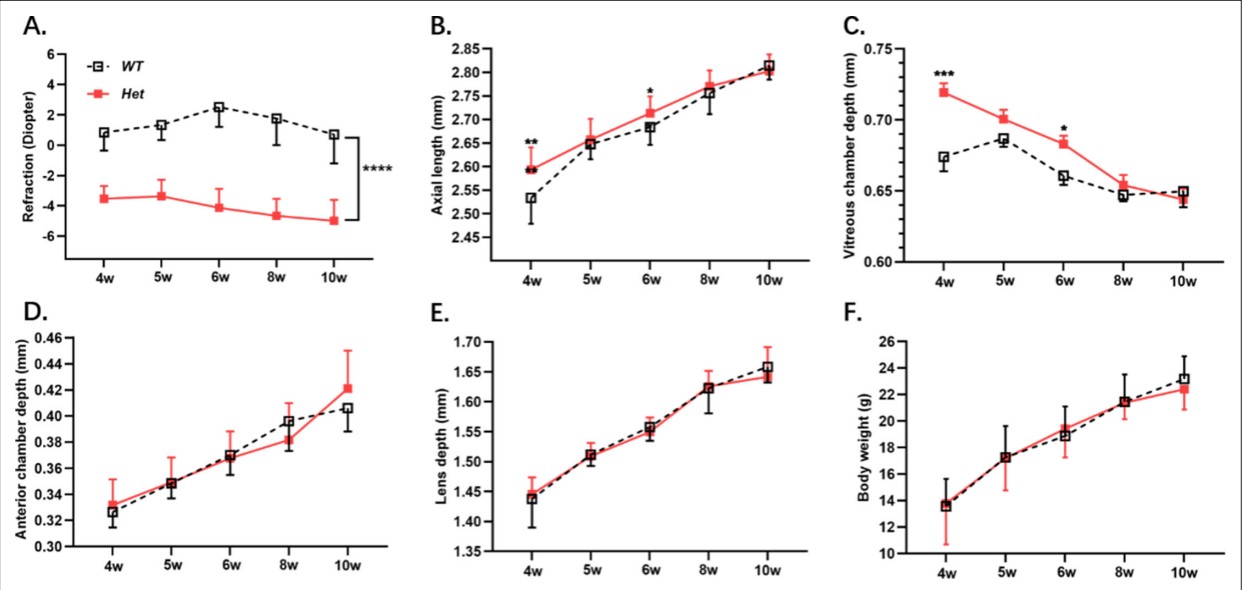

**Figure 2.** *Zc3h11a+/-* mice（n=14）exhibit myopic shifts in refractive parameters compared with WT mice（n=10）. (**A**) *Zc3h11a+/-* mice showed myopia in diopter. (**B**, **C**) Axial length and vitreous chamber depth increased elongation in *Zc3h11a+/-* mice at weeks 4 and 6. (**D–F**) No genotype-dependent differences in anterior chamber depth, lens thickness, or body weight. Statistical significance of differences was assessed using independent-samples t-tests. The results are expressed as mean ± standard deviation (SD), with error bars representing the SD. p-Values are indicated as follows: *p<0.05, **p<0.01, ***p<0.001, and ****p<0.0001.

The online version of this article includes the following figure supplement(s) for figure 2:

**Figure supplement 1.** Generation of *Zc3h11a+/-* mice.

**Figure supplement 2.** Fundus photographs and HE staining of *Zc3h11a+/-* and wild-type (WT) mice at eighth week.

arranged ones (red arrow), with a slight increase in size (*Figure 4G and H*). This suggests impaired light signal capture, transduction, and compromised visual function in *Zc3h11a+/-* mice.

## RNA sequence analysis of molecular and pathways changes in *Zc3h11a+/-* mice retinas

In the retinal transcriptome analysis, 769 genes were differentially expressed (fold change [FC] of at least two and a p-value<0.05) in the *Zc3h11a+/-* group, of which, 303 were upregulated and 466 were downregulated (*Figure 5A*). Gene Ontology (GO) enrichment analysis revealed significant enrichment of differentially expressed genes (DEGs) in the following functions: Zinc ion transmembrane transport (GO:0071577) within metal ion homeostasis, associated with retinal photoreceptor maintenance (*Ugarte and Osborne, 2001*), RNA biosynthesis and metabolism (GO:0006366) in transcriptional regulation, potentially influencing ocular development, negative regulation of NF-κB signaling (GO:0043124) in inflammatory modulation, a pathway involved in scleral remodeling (*Xiao et al., 2025*), calcium ion binding (GO:0005509), critical for phototransduction (*Krizaj and Copenhagen, 2002*), zinc ion transmembrane transporter activity (GO:0005385), participating in retinal zinc homeostasis (*Figure 5B and C*). KEGG pathway enrichment analysis indicated that the DEGs in the *Zc3h11a+/-* group were primarily involved in the PI3K-AKT signaling pathway, MAPK signaling pathway, and neuropsychiatric disease (*Figure 5D*). These findings preliminarily suggest dysregulated NF-κB and PI3K-AKT signaling pathways in the retinas of *Zc3h11a+/-* mice.

## *Zc3h11a* negatively regulates the PI3K-AKT and NF-κB pathways

PI3K-AKT is one of the most important pathways for cell survival, division, autophagy, and differentiation (*Alzahrani, 2019*). Meanwhile, it has been found that *ZC3H11A* can regulate the NF-κB pathway at the level of *IκBα* mRNA export in ZC3-KO cells (*Darweesh et al., 2022*). IκBα is a critical protein that governs NF-κB function predominantly within the cytoplasmic compartment. It plays a pivotal role in inhibiting the activation of NF-κB (*Dyson and Komives, 2012*; *Manavalan et al., 2010*). In

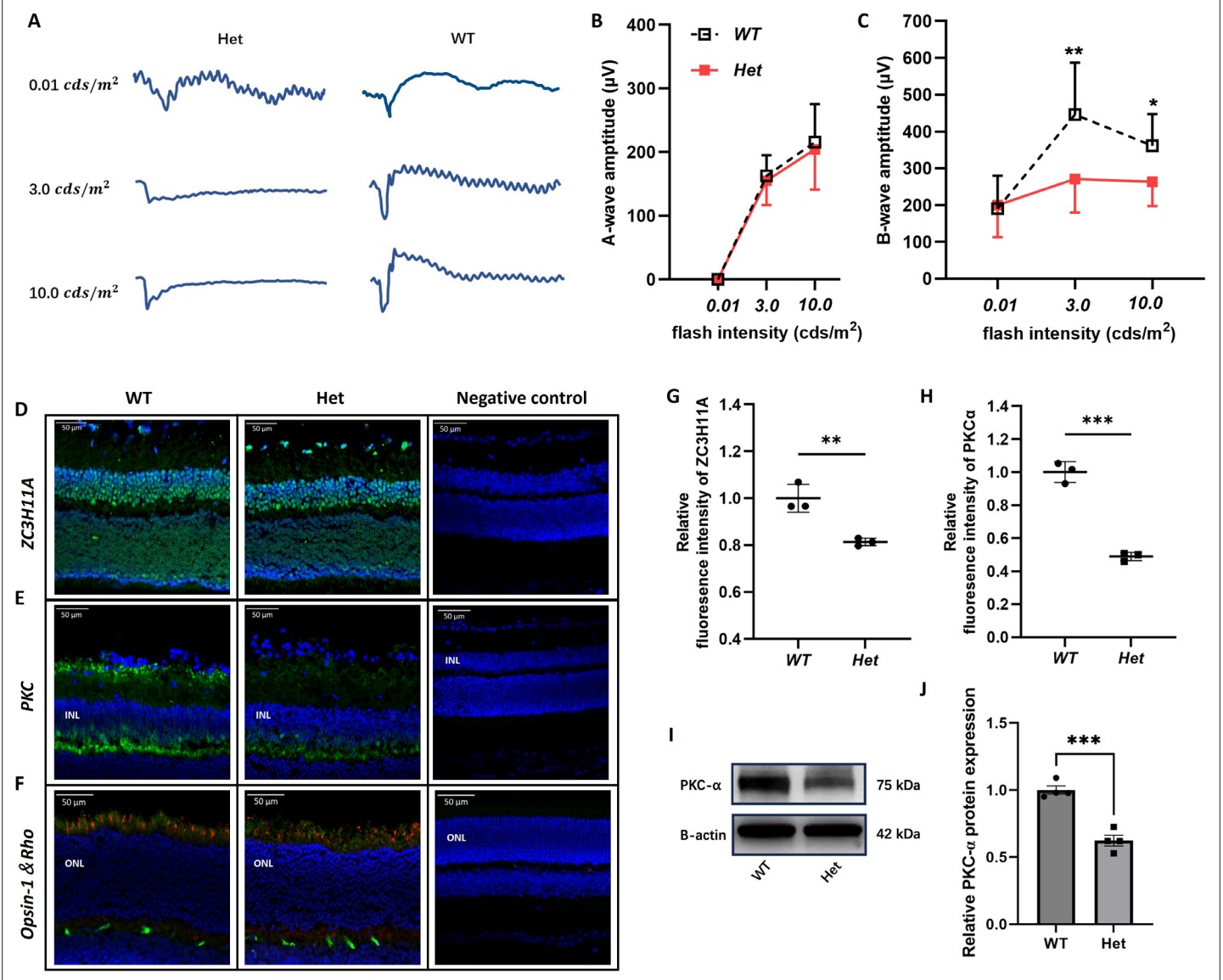

**Figure 3.** *Zc3h11a+/-* mice exhibited bipolar cell dysfunction, accompanied by reduced abundance of the key bipolar cell marker protein PKC-α .
(**A**) Representative scotopic electroretinography (ERG) responses from *Zc3h11a+/-* and wild-type (WT) eyes at dark 0.01 (2.02 log cd·s/m²), dark 3.0 (0.48 log cd·s/m²), and 10.0 (0.98 log cd·s/m²). (**B**, **C**) Quantification of a-wave (photoreceptor function) and b-wave (bipolar cell function) amplitudes. *Zc3h11a+/-* mice (n=12) show significant b-wave reduction at dark 3.0 and dark 10.0 vs WT (n=12). No a-wave differences observed (p>0.1). (**D**–**F**) Immunofluorescence-stained samples to detect Zc3h11a, PKC-α (key bipolar cell marker protein), Opsin-1 (key cone photoreceptors marker protein), and Rhodopsin (key rod photoreceptors marker protein). (**G**, **H**) Zc3h11a and PKC-α protein abundance reduced in the retina of *Zc3h11a+/-* mice（n=3）compared with WT mice（n=3）. (**I**, **J**) Western blot analysis showed that the PKC-α protein content in the retina of *Zc3h11a+/-* mice (n=4) was decreased relative to WT retinas(n=4). Statistical significance of differences was assessed using independent-samples t-tests. The results are expressed as mean ± standard deviation (SD), with error bars representing the SD. p-Values are indicated as follows: *p<0.05, **p<0.01, ***p<0.001, and ****p<0.0001.

The online version of this article includes the following source data and figure supplement(s) for figure 3:

**Source data 1.** Original files used for the western blot analysis in *Figure 3I*.

**Source data 2.** Includes the original western blots for *Figure 3I*, with indicated relevant bands and groupings.

**Figure supplement 1.** Quantitative immunofluorescence analysis of Opsin-1 and Rhodopsin protein levels in the retinas of Zc3h11a+/- and WT mice (n=3 mice/group).

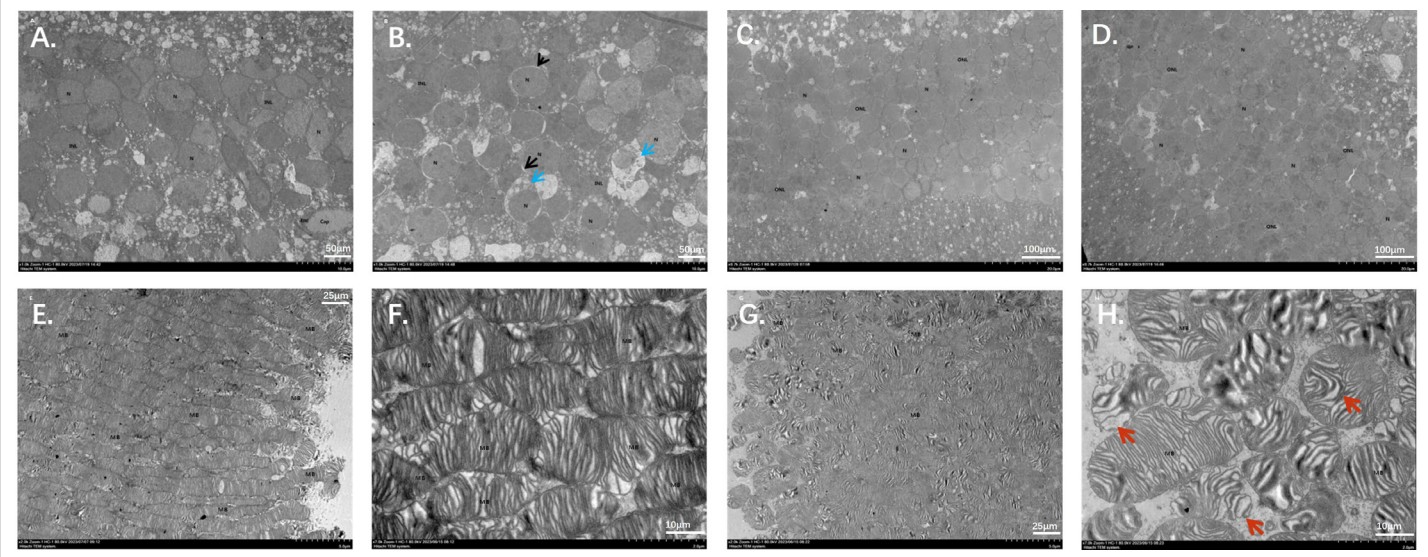

**Figure 4.** Transmission electron microscopy (TEM) showing the retinal abnormalities ultrastructure of *Zc3h11a*[+/-] mice. (**A–B**) Inner nuclear layer (INL). (**A**) Wild-type (WT) mice, normal bipolar cell morphology; (**B**) *Zc3h11a*[+/-] pathological features include enlarged perinuclear gaps (black arrowheads), cytoplasmic edema (blue arrowheads), thinned/lightened cytoplasm in bipolar cells. (**C–D**) Outer nuclear layer (ONL). (**C**) WT mice and (**D**) *Zc3h11a*[+/-] mice, no significant differences in photoreceptor nucleus morphology. (**E–H**) Photoreceptor membrane discs (MB). (**E–F**) WT mice, tightly stacked, uniformly arranged MB. (**G–H**) *Zc3h11a*[+/-] structural disruptions include the outer layer falls off, the local distribution is sparse, and the arrangement is chaotic and loose (red arrow).

addition, AKT promotes IκBα phosphorylation to undergo degradation, which enhances the nuclear translocation of NF-κB (*Alzahrani, 2019*; *Torrealba et al., 2020*). Based on RNA sequencing and the aforementioned literature evidence, this study investigated the regulatory role of *Zc3h11a* in the PI3K-AKT and NF-κB pathways within the mouse retina. To this end, we evaluated the mRNA and protein expression levels of Zc3h11a, PI3K, AKT, p-AKT, IκBα, and NF-κB in the retina. The expression levels of PI3K, p-AKT/AKT, and NF-κB were significantly upregulated, while Zc3h11a and IκBα were downregulated in *Zc3h11a*[+/-] mice (*Figure 6A–L*). Meanwhile, through transfection of overexpression mutant plasmids, it was found that compared to the WT, the mRNA expression levels of *IκBα* in the nucleus of all four mutant types (*ZC3H11A*[V138I], *ZC3H11A*[G43E], *ZC3H11A*[P154L], and *ZC3H11A*[S747T]) were significantly reduced (*Figure 6—figure supplement 1*). These results strongly suggest that *ZC3H11A* likely contributes to myopia pathogenesis by inhibiting both PI3K-AKT and NF-κB signaling pathways.

### *Tgfβ1*, *Mmp2*, and Il6 were increased in *Zc3h11a*[+/-] mice

In animal models of myopia, it has been established that intraocular concentrations of *Tgfβ1* are elevated (*Chen et al., 2013*; *Liu et al., 2022*), especially in the retina and sclera. The abnormal expression of matrix metalloproteinase-2 (*Mmp2*) and interleukin-6 (*Il6*) can induce myopia, and these are regulated by NF-κB (*Libermann and Baltimore, 1990*; *Wu and Schmid-Schönbein, 2011*). qPCR analysis showed increased expression of *Tgfb1*, *Mmp2*, and *Il6* in the sclera and retina of *Zc3h11a*[+/-] mice compared with WT mice (*Figure 7A and B*). The western blot results showed increased expression levels of Tgfβ1 and Mmp2 in the retinas of *Zc3h11a*[+/-] mice (*Figure 7D*). Scleral TEM results suggested that scleral collagen fibers in *Zc3h11a*[+/-] mice were disorganized over a large area with irregular transverse and longitudinal arrangement (*Figure 7C*). These results further corroborate that Zc3h11a[+/-] mice exhibit a myopic phenotype, and that *Zc3h11a* mutation may participate in the development of myopia by inhibiting the NF-κB signaling pathway, thereby upregulating the expression of myopia-related factors such as Mmp2 and Il6.

## Discussion

This study identified and validated a new candidate gene in a large HM cohort, *ZC3H11A*. Moreover, the study provided evidence of the presence of moderate or HM phenotypes and damaged bipolar

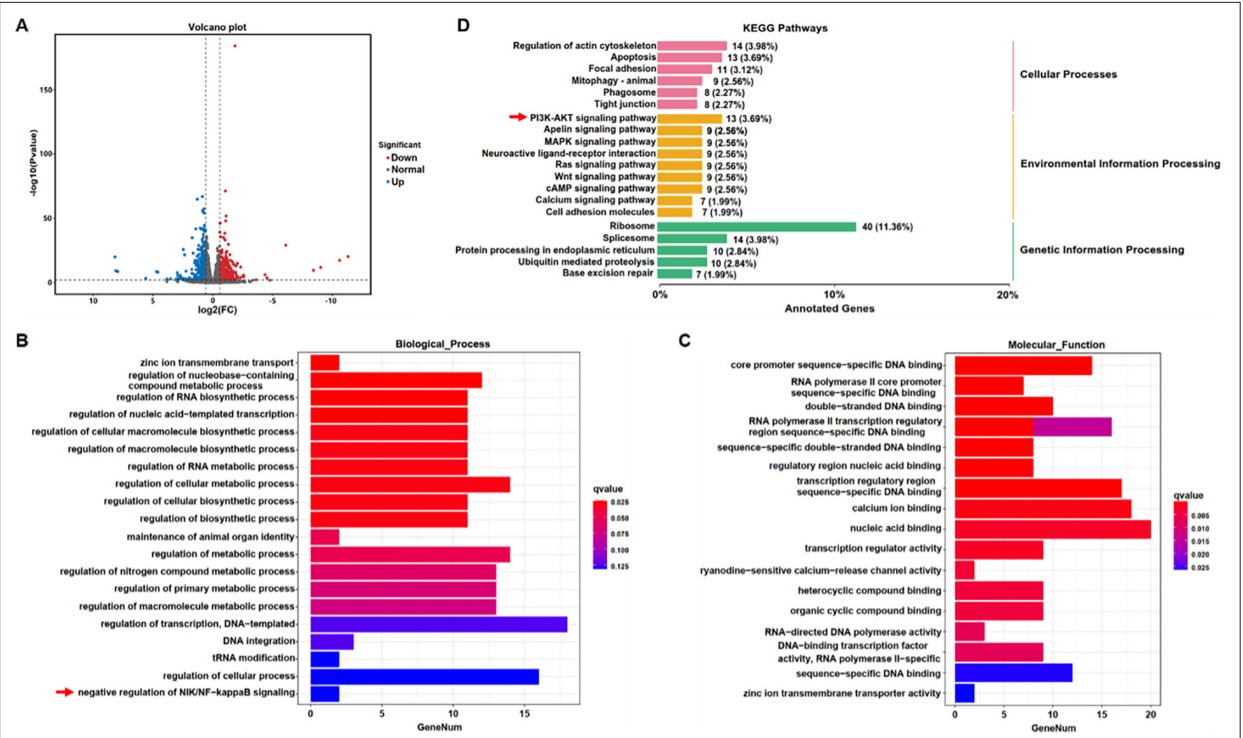

**Figure 5.** RNA sequence reveals pathway dysregulation in the retina of *Zc3h11a*[+/-] mice. (**A**) Volcano plot of differentially expressed genes (DEGs): 769 total (303 upregulated, 466 downregulated in *Zc3h11a*[+/-] (n=3) vs wild-type [WT] (n=3) retinas). (**B**, **C**) Gene Ontology (GO) enrichment analysis: Biological Process: Zinc ion transmembrane transport (GO: 0071577, photoreceptor maintenance), Negative regulation of NF-κB signaling (GO: 0043124, scleral remodeling). Molecular Function: Calcium ion binding (GO: 0005509, phototransduction), Zinc ion transmembrane transporter activity (GO: 0005385, retinal zinc homeostasis). (**D**) KEGG pathway enrichment analysis: Key pathways: PI3K-AKT signaling, MAPK signaling.

cells in vivo by constructing *Zc3h11a*[+/-] mice for the first time. TEM analysis showed ultrastructural changes in parts of both the retina and sclera of *Zc3h11a*[+/-] mice. To understand how *ZC3H11A* regulates the development of HM, retinal transcriptome sequencing was performed. The results revealed that *Zc3h11a* mutation upregulates the PI3K-AKT and NF-κB signaling pathways. In addition, the expression levels of myopia-related factors, such as *Tgfβ1*, *Mmp2,* and *Il6*, were upregulated in the retina and sclera of the *Zc3h11a*[+/-] mice. Therefore, it can be speculated that the ZC3H11A protein may act as a stress-induced nuclear response trigger, its abnormal expression contributing to the early onset of HM.

ZC3H11A is involved in the export and posttranscriptional regulation of selected mRNA transcripts required to maintain metabolic processes in embryonic cells; these are essential for the viability of early mouse embryos (*Younis et al., 2023*). In studies of chickens, mice, and humans, the zinc finger protein ZENK was found to play a role in the development of refractive myopic excursion and lengthening of the ocular axis. Moreover, early growth response gene type 1 *Egr-1* (the human homolog of *ZENK*) activates the *TGF-β1* gene by binding to its promoter, which is thought to be associated with myopia (*Baron et al., 2006*; *Xiao et al., 2022*). Another zinc protein finger protein 644 isoform, *ZNF644*, has recently been identified by WES as causing HM in Han Chinese families (*Shi et al., 2011*). ZC3H11A is also a zinc finger protein, and the findings of the current study revealed that *Zc3h11a*[+/-] mice showed a myopic shift, which is consistent with previous studies reporting reduced protein expression of Zc3h11a in a unilateral induced myopic mouse model (*Fan et al., 2012*).

In vertebrate models, refractive development and ocular axial growth are visually controlled (*Wallman and Winawer, 2004*). The regulation of axial length or refractive error occurs through complex light-dependent retina-to-sclera signaling (*Tkatchenko and Tkatchenko, 2019*). Optical scatter information is processed by the retina and then converted into molecular signals that regulate peripheral retinal growth and scleral connective tissue renewal, ultimately affecting the growth rate of the posterior segment of the eye (*Harper and Summers, 2015*; *Tkatchenko et al., 2006*;

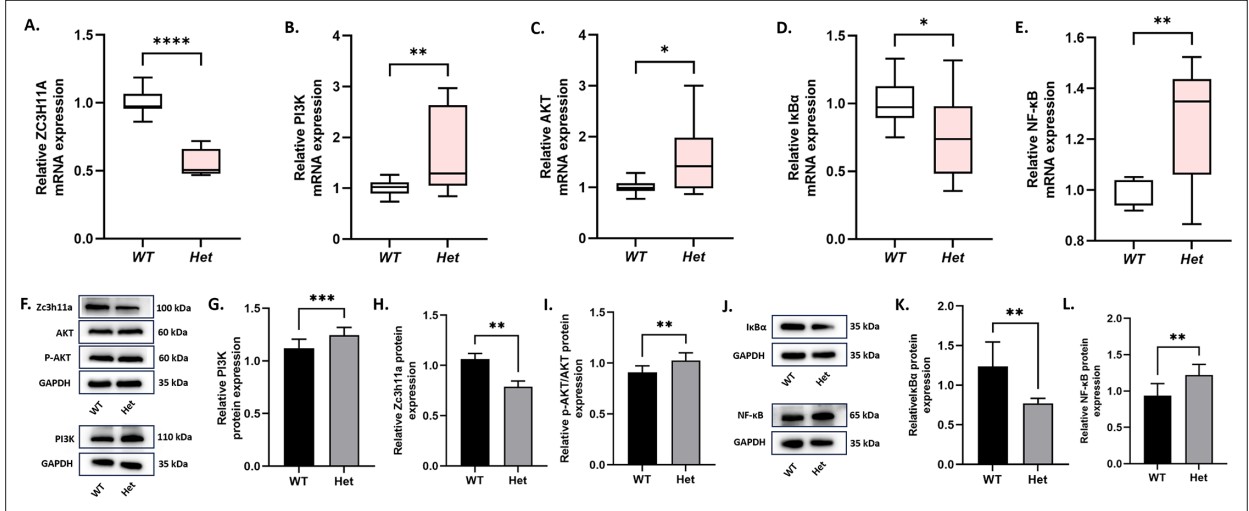

**Figure 6.** Dysregulation of PI3K-AKT and NF-κB signaling pathways in the retinas of *Zc3h11a*[+/-] mice. (**A–E**) qRT-PCR quantification of *ZC3H11A, PI3K, AKT, IκBα*, and *NF-κB* mRNA in the retina (n=3 mice/group): *Zc3h11a* expression was decreased in *Zc3h11a*[+/-] mouse retinas, while *PI3K* and *AKT* expression levels were increased. *IκBα* expression was decreased and *NF-κB* expression was increased. (**F–L**) Western blot analysis of Zc3h11a, PI3K, p-AKT/AKT, IκBα, and NF-κB protein levels in the retina (n=3 mice/group). (**F–I**) Quantitative analysis of Zc3h11a and PI3K levels normalized to GAPDH, while p-AKT levels were normalized to AKT: Zc3h11a expression was decreased in *Zc3h11a*[+/-] mouse retinas, while PI3K and p-AKT/AKT expression levels were increased. (**J–L**) Quantitative analyses of IκBα and NF-κB normalized to GAPDH: IκBα expression was decreased in *Zc3h11a*[+/-] mouse retinas. Statistical significance of differences was assessed using independent-samples t-tests. The results are expressed as mean ± standard deviation (SD), with error bars representing the SD. p-Values are indicated as follows: *p<0.05, **p<0.01, ***p<0.001, and ****p<0.0001.

The online version of this article includes the following source data and figure supplement(s) for figure 6:

**Source data 1.** Original data files for the Zc3h11a western blot analysis presented in *Figure 6F*.

**Source data 2.** Includes the original western blot membrane for Zc3h11a protein in *Figure 6F*, with clearly marked target bands and experimental groupings.

**Source data 3.** Original data files for the PI3K western blot analysis presented in *Figure 6F*.

**Source data 4.** Includes the original western blot membrane for PI3K protein in *Figure 6F*, with clearly marked target bands and experimental groupings.

**Source data 5.** Original data files for the AKT western blot analysis presented in *Figure 6F*.

**Source data 6.** Includes the original western blot membrane for AKT protein in *Figure 6F*, with clearly marked target bands and experimental groupings.

**Source data 7.** Original data files for the p-AKT western blot analysis presented in *Figure 6F*.

**Source data 8.** Includes the original western blot membrane for p-AKT protein in *Figure 6F*, with clearly marked target bands and experimental groupings.

**Source data 9.** Original data files for the IκBα western blot analysis presented in *Figure 6J*.

**Source data 10.** Includes the original western blot membrane for IκBα protein in *Figure 6J*, with clearly marked target bands and experimental groupings.

**Source data 11.** Original data files for the NF-κB western blot analysis presented in *Figure 6J*.

**Source data 12.** Includes the original western blot membrane for NF-κB protein in *Figure 6J*, with clearly marked target bands and experimental groupings.

**Figure supplement 1.** The relative mRNA expression levels of IκBα in the nucleus (n=3/group).

*Tkatchenko et al., 2018*). All cell types of the retina contain myopia-related genes and retinal circuitry driving refractive error. In this study, *Zc3h11a*[+/-] mice were found to have reduced b-wave amplitudes, diminished the abundance of PKC-α, and ultrastructural changes in INL of the retina. Thus, damage to retinal bipolar cells may impair visual signal processing and transduction through neural circuit alterations, thereby contributing to the pathogenesis of HM.

Transcriptome sequencing and quantitative analyses collectively demonstrated downregulated PI3K-AKT and NF-κB signaling pathways in the retinas of *Zc3h11a*[+/-] mice. Given the well-established dysregulation of these pathways in HM (*Lin et al., 2016*), we aimed to investigate how *Zc3h11a*

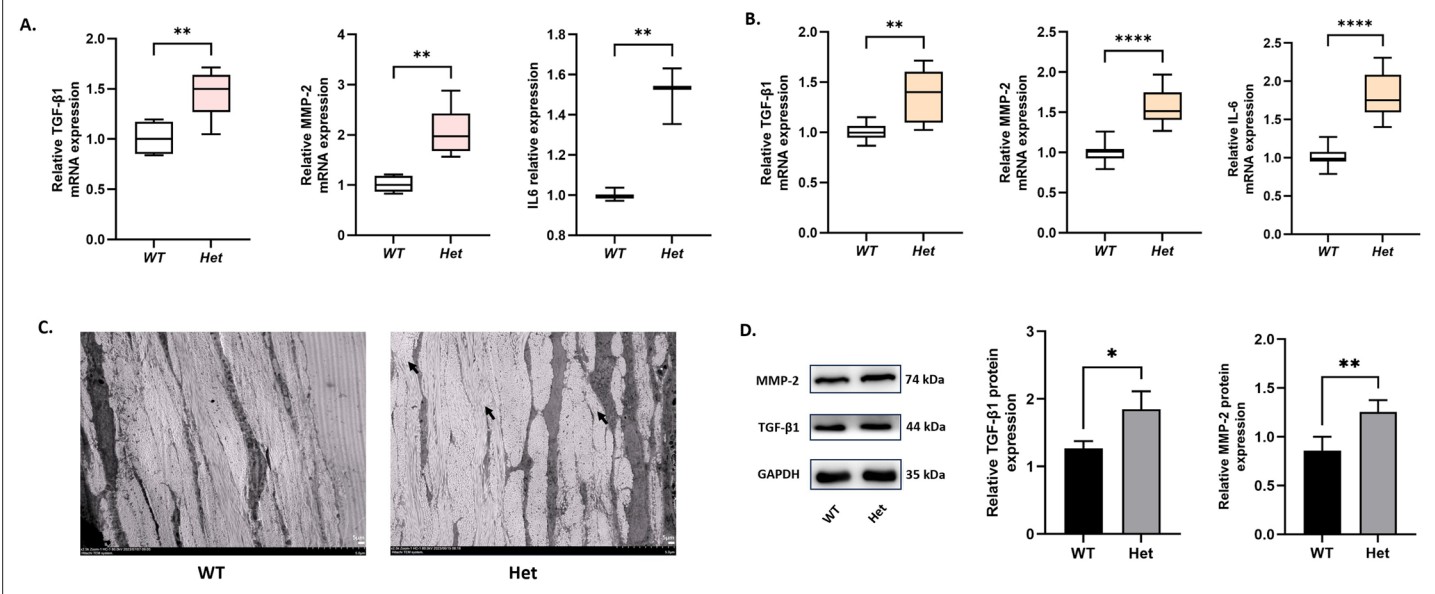

**Figure 7.** Elevated expression of TGF-β1, MMP-2, and IL-6 in retina and sclera of *Zc3h11a*⁺/⁻ mice, with disrupted scleral ultrastructure. (**A**, **B**) qRT-PCR quantification of *Tgf-β1*, *Mmp-2*, and *Il-6* mRNA in retina (**A**) and sclera (**B**) (n=3 mice/group): Expression of *Tgf-β1*, *Mmp-2*, and *Il-6* was increased in both retina and sclera of *Zc3h11a*⁺/⁻ mice. (**C**) Transmission electron microscopy (TEM) structure of sclera, wild-type (WT) mice, organized collagen fibers with regular transverse/longitudinal arrangement. *Zc3h11a*⁺/⁻ mice, disorganized collagen fibers structure with irregular arrangement (black arrows). (**D**) Quantitative analyses of *Tgf-β1* and *Mmp-2* normalized to GAPDH (n=3 mice/group): Expression of *Tgf-β1* and *Mmp-2* was increased in retinas of *Zc3h11a*⁺/⁻ mice. Statistical significance of differences was assessed using independent-samples t-tests. The results are expressed as mean ± standard deviation (SD), with error bars representing the SD. p-Values are indicated as follows: *p<0.05, **p<0.01, ***p<0.001, and ****p<0.0001.

The online version of this article includes the following source data for figure 7:

**Source data 1.** Original data files for the MMP-2 western blot analysis presented in *Figure 7D*.

**Source data 2.** Includes the original western blot membrane for MMP-2 protein in *Figure 7D*, with clearly marked target bands and experimental groupings.

**Source data 3.** Original data files for the TGF-β1 western blot analysis presented in *Figure 7D*.

**Source data 4.** Includes the original western blot membrane for TGF-β1 protein in *Figure 7D*, with clearly marked target bands and experimental groupings.

mediates alterations in these pathways and their potential regulatory cross-talk in the mouse retina. Previous research revealed that a decrease in *Zc3h11a* inhibited the translocation of *IκBα* from the nucleus to the cytoplasm (*Darweesh et al., 2022*). IκBα is a downstream factor of PI3K-AKT and binds to NF-κB (p65) in the cytoplasm, inhibiting its nuclear translocation. The qPCR and western blot results verified that *Zc3h11a* mutation negatively regulates both the PI3K-AKT and NF-κB signaling pathways. Thus, *Zc3h11a* may exert an influence on PI3K-AKT and NF-κB signaling pathways by modulating the cytoplasmic levels of IκBα, contributing to the development of myopia.

NF-κB was first discovered 25 years ago and described as a key regulator of induced gene expression in the immune system, playing a central role in the coordinated control of intrinsic immune and inflammatory responses (*Hayden and Ghosh, 2011*; *Morgan and Liu, 2011*). The downstream factors of this pathway include IL6, IL8, TNF-α, MMP-2, TGF-β1, etc. (*Jimi et al., 2019*; *Yoshida and Whitsett, 2006*). Studies have demonstrated that the MMP-2 and IL6 expression levels are increased in myopic eyes and that inhibiting MMP-2 or IL6 expression will provide some degree of control over myopia progression (*Lin et al., 2016*; *Zhao et al., 2018*). Identification of the PI3K-AKT-NF-κB signaling pathway and downstream myopia effect in our study may open up new opportunities for the prevention and treatment of HM and fundus lesions in the future.

There are a number of limitations to this study that should be acknowledged. First, the patient's ophthalmic examination was not comprehensive enough, and genetic analysis such as segregation analysis was not performed. Second, Zc3h11a homozygous knockout (Homo-KO) mice were not obtained in our study because homozygous deletion of exons confers embryonic lethality (*Younis*

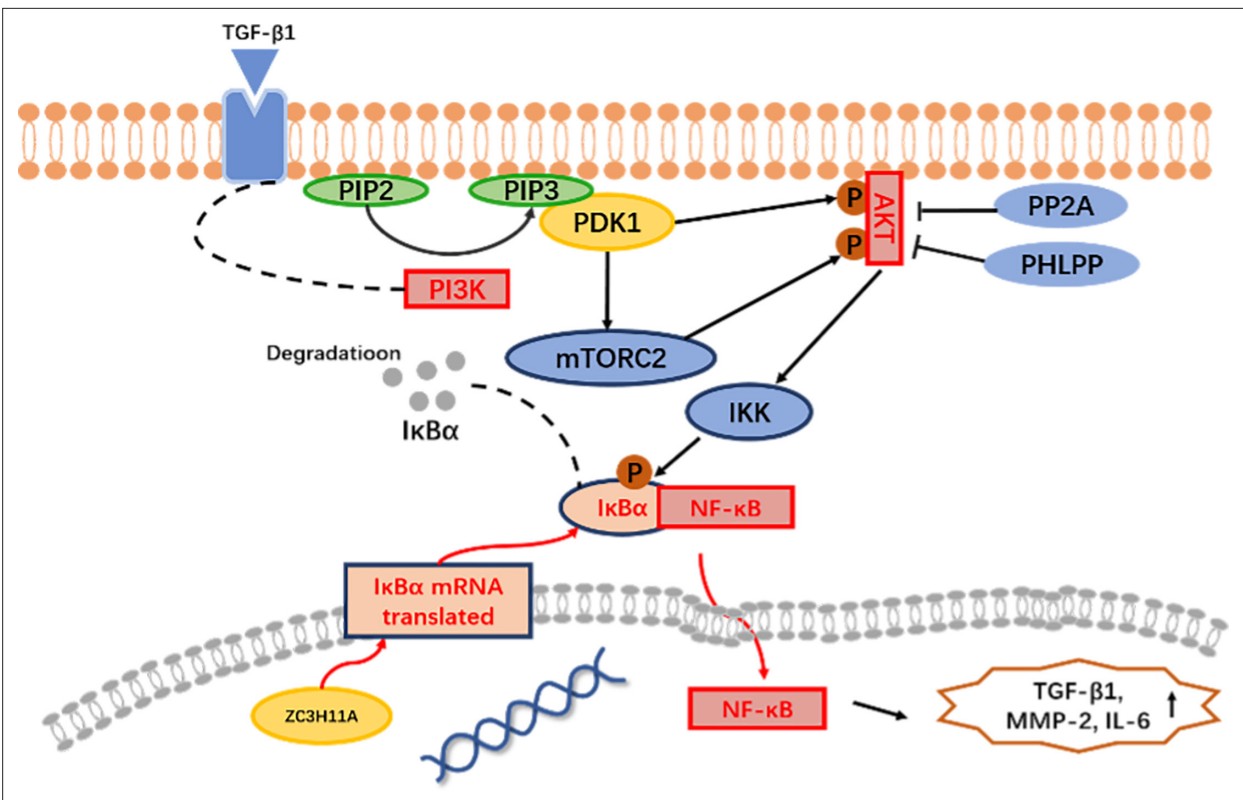

**Figure 8.** Mechanism of *Zc3h11a* haploinsufficiency-driven dysregulation of PI3K-AKT/NF-κB signaling in myopia pathogenesis. Loss of *ZC3H11A* impairs nuclear export of IκBα mRNA, leading to reduced cytoplasmic IκBα protein levels and consequent hyperactivation of NF-κB, and also PI3K-AKT activation. *ZC3H11A* deficiency upregulates PI3K, enhancing the conversion of PIP2 to PIP3, which drives AKT phosphorylation (p-AKT). Activated p-AKT promotes IκBα degradation, further amplifying NF-κB nuclear translocation and transcriptional activity. Downstream, the combined hyperactivity of PI3K-AKT and NF-κB pathways elevates the expression of: TGF-β1 (extracellular matrix remodeling), MMP-2 (collagen degradation), IL-6 (pro-inflammatory signaling), collectively driving scleral thinning and inflammatory responses in myopia pathogenesis.

*et al., 2023*). Third, the axial length of *Zc3h11a*[+/-] mice was longer only at 4 and 6 weeks of age. This may be due to the small size of mouse eyes (1D refractive change corresponds to only 5–6 μm axial length change) (*Schaeffel et al., 2004*) and the theoretical resolution limit of 6 μm of the SD-OCT equipment used in this study (*Zhou et al., 2008b*). Fourth, while the current study observed damage to bipolar cells in *Zc3h11a*[+/-] mice, an in-depth exploration of the underlying mechanism was beyond the scope of the research. Finally, there was a limited observation time in this study, and the effect of *ZC3H11A* on the late refractive system was not assessed.

Overall, this study has provided four key findings. First, it was confirmed that the *ZC3H11A* is a new candidate gene associated with HM in a cohort of Chinese Han individuals. Second, changes in *Zc3h11a* expression were found to affect retinal function, particularly in bipolar cells. Third, changes in *Zc3h11a* expression were found to cause alterations in numerous signaling pathways, the most notable being the PI3K-AKT and NF-κB pathways. Fourth, changes in *Zc3h11a* expression were found to cause changes in the myopia-related genes *Tgfβ1*, *Mmp2*, and *Il6*. The above results suggest that abnormal expression of *Zc3h11a* triggers nuclear translocation defects of *IκBα*, thereby exerting negative feed-back regulation of the PI3K-AKT signaling pathway and inhibiting the NF-κB signaling pathway. At the same time, the increase in Tgfβ1 in myopic eyes stimulates the PI3K-AKT signaling pathway, which leads to activation of the PI3K-AKT signaling pathway, resulting in downstream degradation of IκBα after phosphorylation. This, in turn, increases the nuclear translocation of NF-κB. To sum up, *Zc3h11a* promotes the myopia-associated factor Tgfβ1 by acting directly or indirectly on both the PI3K-AKT and NF-κB signaling pathway-mediated stress reactions and the expression of Mmp2 and Il6, which together, caused the development of early HM (*Figure 8*). Therefore, this model can be used as one of the new strategies for intervention and treatment of HM. However, because the causes of myopia or HM are complex and involve different tissues and molecular pathways in the eye, it is likely that

future research will identify more genes and molecular mechanisms that could provide guidance for clinical intervention and treatment.

## Materials and methods

### Recruitment of subjects

Four sporadic HM patients (aged 15 to 18 years) were recruited from the Eye Hospital of Wenzhou Medical University, three of them came from the Myopia Associated Genetics and Intervention Consortium (MAGIC) project. HM was defined as SE ≤ –6.00 D in either eye (*Flitcroft et al., 2019*). Partial patients underwent certain ophthalmic examinations, including visual acuity (at least three measurements under non-cycloplegia), axial length measurement, and fundus photography.

### WES and the detection of variants

DNA was extracted from all probands using a FlexiGene DNA Kit (QIAGEN, Venlo, The Netherlands), according to the manufacturer's protocol. DNA from all probands underwent WES using a Twist Human Core Exome Kit and an Illumina NovaSeq 6000 sequencing system (150PE) (Berry Genomics Institute, Beijing, China). The mean depth and coverage of the target region were approximately 78.26× and 99.7%, respectively. Sequence reads were aligned to the reference human genome (UCSC hg19) using the Burrows-Wheeler aligner. The variants were called and annotated with Verita Trekker and Enliven software (Transcript ID, ENST00000332127.4, Berry Genomics Institute), respectively.

### *Zc3h11a*$^{+/-}$ mice model construction, husbandry, and ocular biometric measurements

The germline *Zc3h11a*$^{+/-}$ mice were generated by CRISPR/Cas9-mediated gene editing at the embryonic stage on a C57BL/6J background, provided by GemPharmatech Co., Ltd (Nanjing, China). Exons 5–6 of the *Zc3h11a* transcript were recommended as the KO region; this region contains a 244 bp coding sequence. Littermates of *Zc3h11a*$^{+/-}$ and WT mice were used for all experiments. Phenotypic analyses were initiated when the mice reached 4 weeks of age. All animals were housed in the animal husbandry room of Wenzhou Medical University. Mice were housed in standard transparent mouse cages at 22 ± 2°C with a 12-hr light/12-hr dark cycle (light from 8 am to 8 pm, brightness approximately 200–300 lux) and free access to food and water. To exclude potential confounding effects of spontaneous ocular developmental abnormalities, a total of seven mice (four *Zc3h11a*$^{+/-}$ and three WT mice) with small eyes or ocular lesions were excluded from the observation cohort. These anomalies were consistent with the baseline incidence of spontaneous malformations observed in historical colony data of WT C57BL/6J mice (approximately 11%) and were not attributed to the *Zc3h11a* heterozygous KO. Refractive measurements were performed by a researcher blinded to the genotypes. Briefly, in a darkroom, mice were gently restrained by tail-holding on a platform facing an eccentric infrared retinoscope (*Schaeffel et al., 2004*; *Zhou et al., 2008a*). The operator swiftly aligned the mouse position to obtain crisp Purkinje images centered on the pupil using detection software (*Schaeffel et al., 2004*), enabling axial measurements of refractive state and pupil size. Three repeated measurements per eye were averaged for analysis. The anterior chamber depth, lens thickness, vitreous chamber depth, and axial length of the eye were measured by real-time optical coherence tomography (a custom-built OCT) (*Zhou et al., 2008b*). In simple terms, after anesthesia, each mouse was placed in a cylindrical holder on a positioning stage in front of the optical scanning probe. A video monitoring system was used to observe the eyes during the process. Additionally, by detecting the specular reflection on the corneal apex and the posterior lens apex in the two-dimensional OCT image, the optical axis of the mouse eye was aligned with the axis of the probe. Eye dimensions were determined by moving the focal plane with a stepper motor and recording the distance between the interfaces of the eyes. Then, using the designed MATLAB software and appropriate refractive indices, the recorded optical path length was converted into geometric path length. Each eye was scanned three times, and the average value was taken. Body weight, refraction, and ocular biometrics of *Zc3h11a*$^{+/-}$ (n=14) and WT (n=10) mice were assessed at 4, 5, 6, 8, and 10 weeks of age.

## Electroretinography

To evaluate the effect of *Zc3h11a* on the electrophysiological properties of various neuronal populations in the retina, ERG was performed on the right eye of *Zc3h11a*^+/- and WT littermates at 7 weeks of age, at the same time of day (n=12). Both scotopic and photopic ERG responses were evaluated. The mice were dark-adapted overnight, and all procedures were performed under dim red light (<1 lux). The animals were anesthetized with intraperitoneal injection of pentobarbital sodium (40 mg/kg), and the pupils were dilated with 0.5% tropicamide. A heating table (37°C) was used to maintain body temperature. ERG was recorded using a Roland Electrophysiological System (RETI-Port21, Roland Consult, Germany) with ring-shaped corneal electrodes. Scotopic ERG responses were measured under the following parameters: 2.02 log cd·s/m² (dark 0.01): Extremely low light intensity, assessing rod cell function. 0.48 log cd·s/m² (dark 3.0): Moderate light intensity, evaluating the rod-bipolar cell pathway. 0.98 log cd·s/m² (dark 10.0): Higher light intensity, examining bipolar cell responses under strong stimulation. For photopic ERG responses, after 10 min of light adaptation at 25 cd·s/m², recordings were performed at 0.48 log cd·s/m² (light 3.0). Measured parameters included: a-wave amplitude (reflecting photoreceptor function) and b-wave amplitude (indicating bipolar cell activity). The responses of the *Zc3h11a*^+/- mice eyes (right) were compared with those of the WT mice eyes (right).

## Immunofluorescence

Whole mouse eyes (10 weeks, littermates *Zc3h11a*^+/- and WT) were fixed with 4% paraformaldehyde (BL539A, Biosharp, China) for 1 hr and 30% sucrose for dehydration overnight. Then, they were fabric-embedded in a frozen fabric matrix compound at –20°C. Prepared tissue blocks were segmented with a cryostat at a thickness of 12 μm and collected on clean adhesive slides. The slices containing the sections were air-dried at room temperature (RT) for 15 min. After washing 3× with 1× PBS for 5 min, 5% BSA (SW3015, Solarbio, China) and 0.03% Triton X-100 (P0096, Beyotime, China) diluted with 1× PBS were added as permeable membrane-blocking buffers. The slides were incubated for 1 hr at RT in a humid chamber. Then, they were incubated overnight at 4°C with the specific primary antibody Zc3h11a (1:50, 26081, Proteintech, USA). PKC α (1:200, ab32518, Abcam, UK), Opsin-1 (1:200, NB110-74730, Novus Biologicals, USA), and Rhodopsin (1:200, NBP2-25160, Novus Biologicals, USA) were added and the slices were incubated further at 4°C in a humid chamber overnight. After washing 4× with 1× PBS for 6 min, goat anti-rabbit Alexa Fluor 488 (1:500, ab150077, Abcam, UK) was added and the slides were incubated for 90 min at RT. After washing, the slides were mounted with an antifade medium containing DAPI (P0131, Beyotime, China) to visualize the cell nucleus. Sections incubated with 5% BSA and without primary antibodies were used as negative controls. A fluorescence microscope (LSM 880, ZEISS, Germany) was used to examine the slides and capture images. The experiments were repeated in duplicate with three different samples. Fluorescence intensity statistics were analyzed using ZEISS ZEN 3.4 software.

## Transmission electron microscopy

At 10 weeks of age, two mice of each genotype were euthanized, and their eyes were removed. The eyes were fixed in a solution containing 2.5% glutaraldehyde and 0.01 M phosphate buffer (PB) (pH 7.0–7.5) for 15 min while the optic cups were dissected. The optic cups were then fixed with 1% osmium acid at RT away from light for 2 hr, after which they were rinsed three times with 0.1 MPB for 15 min each time. Tissues were sequentially dehydrated in 30%-50%-70%-80%-95%-100%-100% ethanol upstream for 20 min each time, and 100% acetone twice for 15 min each time. Finally, the tissues were embedded in epoxy resin (Polybed 812) mixed 1:1 with acetone. Ultrathin sections were prepared using diamond knives and an EM UC7 ultramicrotome (Leica, Germany), then the sections were stained with 2% aqueous dioxygen acetate and 1% phosphotungstic acid (pH 3.2). Finally, the structures were examined using a transmission electron microscope (HT7700, Hitachi, Tokyo, Japan).

## Cell culture, plasmid construction, and transfection

HEK293T cell line (from Chinese Academy of Sciences, using STR profiling, compared with ExPASy database [defined as CVCL-0063] and testing negative for mycoplasma contamination) was cultured in DMEM containing 10% high-quality 10% fetal bovine serum and 1% penicillin-streptomycin dual antibody. Using restriction endonucleases HindIII and EcoRI to cleave the target gene product (WT,

$ZC3H11A^{V138I}$, $ZC3H11A^{G43E}$, $ZC3H11A^{P154L}$, and $ZC3H11A^{S747T}$) and vector pcDNA3.1 (+), and purify the fragments. Then, under the action of T4 DNA ligase, the target gene was ligated with the vector pcDNA3.1 (+), and the ligated product was transformed into competent bacterial cells. Sequencing was used to identify and select the successfully constructed target gene expression plasmid vector. Finally, a sufficient amount of vector plasmids was obtained through ultrapure endotoxin extraction. Cells were transfected with WT, $ZC3H11A^{V138I}$, $ZC3H11A^{G43E}$, $ZC3H11A^{P154L}$, and $ZC3H11A^{S747T}$ overexpression plasmids using NDE3000 reagent (Western, China).

## RNA sequencing analysis of molecular and pathway changes in the mouse retina

Retinas were harvested from 4-week-old mice for RNA sequencing. Specifically, mice at 4 weeks of age were euthanized via $CO_2$ asphyxiation followed by cervical dislocation. Eyes were immediately enucleated and dissected to isolate intact retinas. Three retinas per group ($Zc3h11a^{+/-}$ and WT) were processed for transcriptomic analysis by Biomarker Biotechnology Co. (Beijing, China). The RNA sequences were mapped to the genome (GRCm38). WebGestalt (http://www.webgestalt.org/) was used to generate GO terms and KEGG pathways.

## RNA extraction and qRT-PCR

To determine the reliability of the transcriptome results, qPCR validation was performed. Total RNA and nuclear RNA were extracted using TRIzol reagent (RC112, Vazyme, China) and Cytodynamic & Nuclear RNA Purification Kit (21000, Norgen Biotek, Canada), respectively, and the purity was confirmed by the OD260/280 nm absorption ratio (1.9–2.1) (NanoDrop 2000, Thermo Scientific, USA). Total RNA (2 μg) was reverse-transcribed to cDNA using a cDNA Synthesis Kit (R323, Vazyme, China). qPCR was performed with a RT-PCR detection system (Applied Biosystems, CA, USA) using a SYBR Premix Ex Taq Kit (Q711, Vazyme, China), according to the manufacturer's instructions. qRT-PCR was performed (ABI-Q6, CA, USA) in a 20 μL reaction, under the following conditions: 95°C for 10 min, followed by 40 cycles of amplification at 95°C for 10 s and 60°C for 60 s. Melting curve analysis was used to determine specific amplification. All experiments were performed in triplicate. Relative quantification was performed using the ΔΔCt method. The specific gene products were amplified using the following primer pairs (*Supplementary file 1*).

## Western blot

The mouse retina was separated immediately after enucleation of the eyeball and lysed in RIPA (10 mM Tris-Cl, 100 mM NaCl, 1 mM EDTA, 1 mM NaF, 20 mM $Na_4P_2O_7$, 2 mM $Na_3VO_4$, 1% Triton X-100, 10% glycerol, 0.1% sodium dodecyl sulfate, and 0.5% deoxycholate) lysis buffer containing protease and phosphatase inhibitors (P1045, Beyotime, China). Equal amounts of protein (15 μg) were separated on 10% Tris-glycine gel, transferred to polyvinylidene fluoride (PVDF) membranes, and blocked with 5% skim milk. The primary antibodies used included ZC3H11A (ab99930, Abcam, UK), AKT (#4691, Cell Signaling Technology, USA), p-AKT (#4060, Cell Signaling Technology, USA), IκBα (ab32518, Abcam, UK), NF-κB (ab32536, Abcam, UK), TGF-β1 (21898, Proteintech, USA), PKC-α (ab32518, Abcam, UK), and MMP-2 (ab92536, Abcam, UK) diluted to 1:1000 in TBST-5% milk. Then, the membranes were incubated with goat anti-rabbit IgG conjugated with HRP (1:2000 in TBST-5% milk) (SA00001-2, Proteintech, USA) for 90 min at RT and developed using western blotting reagents (BL523B, Bioshark, China). GAPDH (AF2823, Byotime, China) was used as the internal control.

## Software and statistical analysis

All experiments were repeated at least once, and sample sizes and reported results reflect the cumulative data for all trials of each experiment. Each result is expressed as the mean ± standard deviation (SD). Pearson's test was used to assess the normality of the data. The normally distributed data were subjected to parametric analyses. Unpaired Student's t-tests were used for parametric analyses between two groups. Data that were not normally distributed or had a sample size that was too small were subjected to nonparametric analyses. Nonparametric tests between two groups were performed using the Wilcoxon signed-rank test for matched pairs, the Mann-Whitney U-test, or the Kruskal-Wallis test for multiple comparisons (GraphPad Prism 9, La Jolla, CA, USA). A difference was considered

statistically significant when the p-value was less than 0.05 and highly significant if it was less than 0.01 or less than 0.001.

## Acknowledgements

The authors wish to thank all of the members of our lab. Additionally, we thank the myopia and big data research group at Eye Hospital of Wenzhou Medical University for their technical assistance. This work was supported by the National Natural Science Foundation of China (82101176), the Natural Science Foundation of Zhejiang Province (LTGD23H120002), the Health Technology Plan Project in Zhejiang Province (2023KY151), and the Science and Technology Project of Wenzhou (Y20220774).

## Additional information

### Funding

| Funder | Grant reference number | Author |
| --- | --- | --- |
| National Natural Science Foundation of China | 82101176 | Xinting Liu |
| Natural Science Foundation of Zhejiang Province | LTGD23H120002 | Xinting Liu |
| Health Technology Plan Project in Zhejiang Province | 2023KY151 | Chong Chen |
| Science and Technology Project of Wenzhou | Y20220774 | Chong Chen |

The funders had no role in study design, data collection and interpretation, or the decision to submit the work for publication.

### Author contributions

Chong Chen, Conceptualization, Resources, Funding acquisition, Validation, Writing – original draft, Writing – review and editing; Qian Liu, Data curation, Software, Formal analysis, Methodology, Writing – original draft; Cheng Tang, Software, Validation, Investigation, Methodology, Writing – original draft; Yu Rong, Software, Validation, Methodology; Xinyi Zhao, Dandan Li, Software, Methodology; Fan Lu, Resources, Data curation, Investigation, Writing – review and editing; Jia Qu, Conceptualization, Resources, Writing – review and editing; Xinting Liu, Conceptualization, Resources, Investigation, Writing – original draft, Project administration, Writing – review and editing

### Author ORCIDs

Chong Chen ⬚ https://orcid.org/0009-0007-0620-5997
Xinting Liu ⬚ https://orcid.org/0000-0002-2798-448X

### Ethics

Human subjects: The studies that involved human participants were approved by the Eye Hospital, Wenzhou Medical University (Permit Number: 2021-015-K-12-01), and were carried out in strict adherence to the guidelines of the Helsinki Declaration. Informed consent was obtained from each subject. All animal experiments were approved by the Animal Care and Ethics Committee at Wenzhou Medical University (Permit Number: YSG23103011) and were performed according to the guidelines set forth in the Association for Research in Vision and Ophthalmology Statement for the Use of Animals in Ophthalmic and Visual Research.

Reviewer #1 (Public Review): https://doi.org/10.7554/eLife.91289.4.sa1
Author response https://doi.org/10.7554/eLife.91289.4.sa2

# Additional files

## Supplementary files
Supplementary file 1. Sequence of oligonucleotides.

MDAR checklist

## Data availability
The sequencing datasets reported in this paper is available at GSA (https://ngdc.cncb.ac.cn/gsub) with accession number: CRA028295.

The following dataset was generated:

| Author(s) | Year | Dataset title | Dataset URL | Database and Identifier |
|---|---|---|---|---|
| Chen C | 2025 | Genomics Study on High Myopia (ZC3H11A) | https://ngdc.cncb.ac.cn/gsa/browse/CRA028295 | Genome Sequence Archive, CRA028295 |

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
