## [Editor Report · eLife Assessment]

This work investigates ZC3H11A as a cause of high myopia through the analysis of human data and experiments with genetic knockout of Zc3h11a in mouse, providing a **useful** model of myopia. The evidence supporting the conclusion is still **incomplete** in the revised manuscript as the concerns raised in the previous review were not fully addressed. The article would benefit from a more robust genetic analysis and comprehensive presentation of human phenotypic data to clarify the modes of inheritance in the families, currently limited by loss of patient follow-up and addressing whether there is a reduction in bipolar cell number or decreased marker protein expression through cell counts or quantifiable, less saturated Western blots. The work will be of interest to ophthalmologists and researchers working on myopia

---

## [Referee Report · Reviewer #1 (Public Review)]

The authors reported that mutations were identified in the ZC3H11A gene in four adolescents from 1015 high myopia subjects in their myopia cohort. They further generated Zc3h11a knockout mice utilizing the CRISPR/Cas9 technology.

The main claims are only partially supported. The reviewers still have the concerns of (1) the modes of inheritance for the families need to be shown; (2) the phenotype of heterozygous mutant mice is too weak; (3) the authors still have not addressed the biological question of whether there are fewer bipolar cells or decreased expression of the marker protein. This would involve counting cells, which they have not done. The blots they show do not appear to support their quantifications. Considering the sensitivity of quantifying nearly saturated blots, the authors should show blots that are not exposed to that level of saturation.

---

## [Author Response]

The following is the authors’ response to the previous reviews.

**Reviewer #2 (Public review):**
Summary:The authors reported that mutations were identified in the ZC3H11A gene in four adolescents from 1015 high myopia subjects in their myopia cohort. They further generated Zc3h11a knockout mice utilizing the CRISPR/Cas9 technology.Comments on revisions:Chong Chen and colleagues revised the manuscript; however, none of my suggestions from the initial review have been sufficiently addressed.(1) I indicated that the pathogenicity and novelty of the mutation need to be determined according to established guidelines and databases. However, the conclusion was still drawn without sufficient justification.

Thank you for your valuable feedback on the assessment of mutation pathogenicity and novelty. We regret to inform you that complete familial genetic information required for segregation analysis is currently unavailable in this study. Despite our exhaustive efforts to contact the four mutation carriers and their relatives, we encountered the following uncontrollable limitations: Two patients could not be further traced due to invalid contact information, one patient had relocated to another region, making sample collection logistically unfeasible, the remaining patient explicitly declined family participation in genetic testing due to privacy concerns.

We fully acknowledge that the lack of pedigree data may affect the certainty of pathogenicity evaluation. To address this limitation, we systematically analyzed the four *ZC3H11A* missense mutations (c.412G>A p.V138I, c.128G>A p.G43E, c.461C>T p.P154L, and c.2239T>A p.S747T) based on ACMG guidelines and database evidence. The key findings are summarized below: All of the identified mutations exhibited very low frequencies or does not exist in the Genome Aggregation Database (gnomAD) and Clinvar, and using pathogenicity prediction software SIFT, PolyPhen2, and CADD, most of them display high pathogenicity levels. Among them, c.412G>A, c.128G>A and c.461C>T were located in or around a domain named zf-CCCH_3 (Figure 1A and B). Furthermore, all of the mutation sites were located in highly conserved amino acids across different species (Figure 1C). The four mutations induced higher structural flexibility and altered the negative charge at corresponding sites, potentially disrupting protein-RNA interactions (Figure 1D and E). Concurrently, overexpression of mutant constructs (*ZC3H11A*-V138I, *ZC3H11A*-G43E, *ZC3H11A*-P154L, and *ZC3H11A*-S747T) revealed significantly reduced nuclear IκBα mRNA levels compared to the wild-type, suggesting impaired NF-κB pathway regulation (Supplementary Figure 4). Zc3h11a knockout mice also exhibited a myopic phenotype, with alterations in the PI3K-AKT and NF-κB signaling pathways. Integrating this evidence, the mutations meet the following ACMG criteria: PM1 (domain-located mutations), PM2 (extremely low population frequency), PP3 (computational predictions supporting pathogenicity), PS3 (functional validation via experimental assays). Under the ACMG framework, these mutations are classified as "Likely Pathogenic".

Regarding the novelty of this mutation, comprehensive searches in ClinVar, dbSNP, and HGMD databases revealed no prior reports associating this variant with myopia. Similarly, a PubMed literature search identified no direct evidence linking this mutation to myopia. Based on this evidence, we classify this variant as a likely pathogenic and novel mutation.

On the other hand, we acknowledge that the absence of family segregation data may reduce the confidence in pathogenicity assessment. Nevertheless, functional experiments and converging multi-level evidence strongly support the reliability of our conclusion. Future studies will prioritize family-based validation to strengthen the evidence chain. We sincerely appreciate your attention to this matter and kindly request your understanding of the practical limitations inherent to this research.

(2) The phenotype of heterozygous mutant mice is too weak to support the gene's contribution to high myopia. The revised manuscript does not adequately address these discrepancies. Furthermore, no explanation was provided for why conditional gene deletion was not used to avoid embryonic lethality, nor was there any discussion on tissue- or cell-specific mechanistic investigations.

We sincerely appreciate your insightful comments regarding the relationship between murine phenotypes and human disease. We fully acknowledge your concerns about the phenotypic strength of *Zc3h11a* heterozygous mutant mice and their association with high myopia (HM) pathogenesis. Here we provide point-by-point responses to your valuable comments: Our study demonstrates that Zc3h11a heterozygous mutant mice exhibit myopic refractive phenotypes with upregulated myopia-associated factors (TGF-β1, MMP2, and IL6), although axial elongation did not reach statistical significance. Notably, at 4 and 6 weeks of age, Het mice did display longer axial lengths and vitreous chamber depths compared to WT mice. While these differences did not reach statistical significance at other time points, an increasing trend was still observed. Several technical considerations may explain these findings: The small murine eye size (where 1D refractive change corresponds to only 5-6μm axial length change). The theoretical resolution limit of 6μm for the SD-OCT device used in this study. These factors likely contributed to the marginal statistical significance observed in the subtle changes of vitreous chamber depth and axial length measurements. Additionally, existing research indicates that axial length measurements from frozen sections in age-matched mice tend to be longer than those obtained through in vivo measurements. This phenomenon may reflect species differences between humans and mice - while both show significant refractive power changes, the axial length differences are less pronounced in mice. These results align with previous reports of phenotypic differences between mouse models and human myopia.

To address these issues comprehensively, we have added a dedicated discussion section in the revised manuscript specifically examining these axial length measurement considerations, following your valuable suggestion.

Additionally, we regret to inform you that the currently available floxed ZC3H11A mouse strain requires a minimum of 12-18 months for custom construction, which exceeds our research timeline due to current resource limitations in our team. To address this gap, we have supplemented the discussion section with additional content regarding tissue- and cell-specific mechanisms. Based on your constructive suggestions, we will prioritize the following in our subsequent work: Collaborate with transgenic animal centers to generate *Zc3h11a* conditional knockout mice. Evaluate the impact of specific knockouts on myopia progression using form-deprivation (FDM) models. While we recognize the limitations of our current study, we believe that by integrating clinical cohort data, phenotypic evidence, and functional experiments, this research provides valuable directional evidence for *ZC3H11A*'s potential role in myopia pathogenesis. Your comments will significantly contribute to improving our future research design, and we sincerely hope you can recognize the exploratory significance of our current findings.

(3) The title, abstract, and main text continue to misrepresent the role of the inflammatory intracellular PI3K-AKT and NF-κB signaling cascade in inducing high myopia. No specific cell types have been identified as contributors to the phenotype. The mice did not develop high myopia, and no relationship between intracellular signaling and myopia progression has been demonstrated in this study.

Thank you for your valuable comments regarding the interpretation of signaling pathways in our study. We fully acknowledge your rigorous concerns about the role of PI3K-AKT and NF-κB signaling cascades in high myopia and recognize that we did not identify specific cell types contributing to the observed phenotype. In response to your feedback, we have removed the hypothetical statement linking genetic changes within inflammatory cells to the development of myopia. The current interpretation is strictly based on experimental evidence of pathway relevance and is supported by the theoretical basis presented in the reference, specifically that loss of Zc3h11a leads to activation of the PI3K-AKT and NF-κB pathways in retinal cells, contributing to the myopic phenotype.

**Author response image 1. sa2fig1:** 

Model of the association between inflammation and myopia progression. Activated mAChR3 (M3R) activates phosphoinositide 3-kinase (PI3K)–AKT and mitogen-associated protein kinase (MAPK) signaling pathways, in turn activating NF-κB and AP1 (i.e., the Jun.-Fos heterodimer) and stimulating the expression of the target genes NF-κB, MMP2, TGFβ, IL- 1β and -6, and TNF-α. MMP2 and TGF-β promote tissue remodeling and TNF-α may act in a paracrine feedback loop in the retina or sclera to activate NF-κB during myopia progression.

To address the limitations raised, we will prioritize the following in future studies: Cell-type-specific knockout models to identify key cellular contributors. Mechanistic investigations to establish causal relationships between signaling pathways and myopia progression. We sincerely appreciate your rigorous review, which has significantly improved the scientific accuracy and clarity of our manuscript. We believe the revised version better reflects both the novelty and limitations of our findings. We kindly request your recognition of the study’s contributions while acknowledging its current constraints.

**Reviewer #3 (Public review):**
Chen et al have identified a new candidate gene for high myopia, ZC3H11A, and using a knock-out mouse model, have attempted to validate it as a myopia gene and explain a potential mechanism. They identified 4 heterozygous missense variants in highly myopic teenagers. These variants are in conserved regions of the protein, and predicted to be damaging, but the only evidence the authors provide that these specific variants affect protein function is a supplement figure showing decreased levels of IκBα after transfection with overexpression plasmids (not specified what type of cells were transfected). This does not prove that these mutations cause loss of function, in fact it implies they have a gain-of-function mechanism. They then created a knock-out mouse. Heterozygotes show myopia at all ages examined but increased axial length only at very early ages. Unfortunately, the authors do not address this point or examine corneal structure in these animals. They show that the mice have decreased B-wave amplitude on electroretinogram (a sign of retinal dysfunction associated with bipolar cells), and decreased expression of a bipolar cell marker, PKCα. On electron microscopy, there are morphologic differences in the outer nuclear layer (where bipolar, amacrine, and horizontal cell bodies reside). Transcriptome analysis identified over 700 differentially expressed genes. The authors chose to focus on the PI3K-AKT and NF-κB signaling pathways and show changes in expression of genes and proteins in those pathways, including PI3K, AKT, IκBα, NF-κB, TGF-β1, MMP-2 and IL-6, although there is very high variability between animals. They propose that myopia may develop in these animals either as a result of visual abnormality (decreased bipolar cell function in the retina) or by alteration of NF-κB signaling. These data provide an interesting new candidate variant for development of high myopia, and provide additional data that MMP2 and IL6 have a role in myopia development. For this revision, none of my previous suggestions have been addressed.
**Reviewer #3 (Recommendations for the authors):**
None of these suggestions were addressed in the revision:Major issues:(1) Figure 2: refraction is more myopic but axial length is not longer - why is this not discussed and explored? The text claims the axial length is longer, but that is not supported by the figure. If this is a measurement issue, that needs to be discussed in the text.

We sincerely appreciate your valuable comments regarding the relationship between refractive status and axial length in our study. In response to your concerns, we have conducted an in-depth analysis and would like to address the issues as follows:

Our data demonstrate significant differences in refractive error between heterozygous (Het) and wild-type (WT) mice during the 4-10 weeks. Notably, at 4 and 6 weeks of age, Het mice did exhibit longer axial lengths and greater vitreous chamber depth compared to WT mice, although these differences did not reach statistical significance at other time points while still showing an increasing trend. Additional measurements of corneal curvature revealed no significant differences between groups. Considering the small size of mouse eyes (where a 1D refractive change corresponds to only 5-6μm axial length change) and the theoretical resolution limit of 6μm for the SD-OCT device used in this study, these technical factors may account for the marginal statistical significance of the observed small changes in vitreous chamber depth and axial length measurements. Furthermore, existing studies have shown that axial length measurements from frozen sections tend to be longer than those obtained from in vivo measurements in age-matched mice. These considerations provide plausible explanations for the apparent discrepancy between refractive changes and axial length parameters. Following your suggestion, we have added a dedicated discussion section addressing these axial length measurement issues in the revised manuscript. We fully understand your concerns regarding data consistency, and your comments have prompted us to conduct more comprehensive and thorough analysis of our results. We believe the revised manuscript now more accurately reflects our findings while providing important technical references for future studies.

(2) Slipped into the methods is a statement that mice with small eyes or ocular lesions were excluded. How many mice were excluded? Are the authors ignoring another phenotype of these mice?

We appreciate your attention to the exclusion criteria and their implications. Below we provide a detailed clarification: A total of 7 mice (4 Het-KO and 3 WT) with small eyes or ocular lesions were excluded from the observation cohort. These anomalies were consistent with the baseline incidence of spontaneous malformations observed in historical colony data of wild-type C57BL/6J mice (approximately 11%), and were not attributed to the *Zc3h11a* heterozygous knockout. We have added the above content in the methods section. Your insightful comment has significantly strengthened our reporting rigor. We hope this clarification alleviates your concerns regarding potential selection bias or overlooked phenotypes.

Minor/Word choice issues:All the figure legends need to be improved so that each figure can be interpreted without having to refer to the text.

Thank you for your valuable comments. We have made modifications to the legend of each graphic, as detailed in the main text.

Abstract: line 24: use refraction, not "vision"

Thank you for your valuable comments. The “Vision” has been changed to “refraction”.

Line 28: re-word "density of bipolar cell-labeled proteins" Do the authors mean density of bipolar cells? Or certain proteins were less abundant in bipolar cells?

Thank you for your rigorous review of this terminology. We acknowledge the need to clarify the precise meaning of the phrase "density of bipolar cell-labeled proteins." In the original text, this term specifically refers to the expression abundance of the bipolar cell-specific marker protein PKCα, which was identified using immunofluorescence labeling techniques. Specifically: We utilized PKCα (a bipolar cell marker) to label bipolar cell populations. The "density" was quantified by measuring the fluorescence signal intensity per unit area in confocal microscopy images, rather than direct cell counting. This metric reflects changes in the expression of the specific marker protein (PKCα) within bipolar cells, which indirectly correlates with alterations in bipolar cell populations. To address ambiguity, we have revised the terminology throughout the manuscript to "bipolar cell-labelled protein PKCα immunofluorescence abundance".

Additionally, since fluorescence intensity quantification is inherently semi-quantitative, we have included Western blot results for PKCα in the revised manuscript (Figure 3I, J) to validate the expression changes observed via immunofluorescence. We sincerely appreciate your feedback, which has significantly improved the precision of our manuscript.

Line 45: axial length, not ocular axis

Thank you for your valuable comments. The “ocular axis” has been changed to “axial length”.

Lines73-75: confusing

Thank you for your valuable comments. The relevant content has been modified to “Multiple zinc finger protein genes (e.g., *ZNF644, ZC3H11B, ZFP161, ZENK*) are associated with myopia or HM. Of these, *ZC3H11B* (a human homolog of *ZC3H11A*) and five GWAS loci (Schippert et al., 2007; Shi et al., 2011; Szczerkowska et al., 2019; Tang et al., 2020; Wang et al., 2004) correlate with AL elongation or HM severity. Proteomic studies further suggest ZC3H11A involvement in the TREX complex, implicating RNA export mechanisms in myopia pathogenesis”

Line 138: what is dark 3.0 and dark 10.0

Thank you for your valuable comments. The relevant content has been modified to “Upon dark adaptation, b-wave amplitudes in seven-week-old Het-KO mice were significantly lower at dark 3.0 (0.48 log cd·s/m²) and dark 10.0 (0.98 log cd·s/m²) compared to WT mice.” A detailed description has been added to the main text methods.

Line 171-175: the GO terms of "biological processes" and "molecular functions" are so broad as to be meaningless.

Thank you for your valuable comments. The relevant content has been modified to “GO enrichment analysis revealed significant enrichment of differentially expressed genes in the following functions: Zinc ion transmembrane transport (GO:0071577) within metal ion homeostasis, associated with retinal photoreceptor maintenance (Ugarte and Osborne, 2001), RNA biosynthesis and metabolism (GO:0006366) in transcriptional regulation, potentially influencing ocular development, negative regulation of NF-κB signaling (GO:0043124) in inflammatory modulation, a pathway involved in scleral remodelling (Xiao et al., 2025), calcium ion binding (GO:0005509), critical for phototransduction (Krizaj and Copenhagen, 2002), zinc ion transmembrane transporter activity (GO:0005385), participating in retinal zinc homeostasis (Figure 5C and D).”

Line 257-259: which results indicated loss of Zc3h11a inhibited translocation of IκBα from nucleus to cytoplasm? Results of this study, or the previously referenced study?

We sincerely appreciate your critical inquiry regarding the mechanistic relationship between *Zc3h11a* deficiency and IκBα translocation. We are grateful for this opportunity to clarify this important point. The findings regarding *Zc3h11a*-mediated regulation of IκBα mRNA nuclear export and its impact on NF-κB signaling originate from the study by Darweesh et al. The key experimental evidence demonstrates that: The depletion of *Zc3h11a* leads to nuclear retention of IκBα mRNA, resulting in failure to maintain normal levels of cytoplasmic IκBα mRNA and protein. This defect in IκBα mRNA export disrupts the essential inhibitory feedback loop on NF-κB activity, causing hyperactivation of this pathway. This manifests as upregulation of numerous innate immune-related mRNAs, including IL-6 and a large group of interferon-stimulated genes.While our study references this mechanism to explain the observed NF-κB dysregulation in *Zc3h11a* Het-KO mice, the specific nuclear export mechanism was indeed elucidated by Darweesh et al. The reference has been inserted into the corresponding position in the main text. Importantly, our research extends these previous molecular insights into the phenotypic context of myopia.

We sincerely regret any ambiguity in the original text and deeply appreciate your rigorous approach in ensuring proper attribution of these fundamental findings. Your comment has significantly improved the clarity and accuracy of our manuscript.

Figure 6 shows decrease of both mRNA and protein expression, but nothing about translocation.

Thank you for your valuable comments. The research results of Darweesh et al. showed that Zc3h11a protein plays a role in regulation of NF-κB signal transduction. Depletion of Zc3h11a resulted in enhanced NF-κB mediated signaling, with upregulation of numerous innate immune related mRNAs, including IL-6 and a large group of interferon-stimulated genes. IL-6 upregulation in the absence of the Zc3h11a protein correlated with an increased NF-κB transcription factor binding to the IL-6 promoter and decreased IL-6 mRNA decay. The enhanced NF-κB signaling pathway in *Zc3h11a* deficient cells correlated with a defect in IκBα inhibitory mRNA and protein accumulation. Upon *Zc3h11a* depletion The IκBα mRNA was retained in the cell nucleus resulting in failure to maintain normal levels of the cytoplasmic IκBα mRNA and protein that is essential for its inhibitory feedback loop on NF-κB activity. These findings demonstrate that ZC3H11A can regulate the NF-κB pathway by controlling the translocation of IκBα mRNA, a mechanism that was indeed elucidated by Darweesh et al. We sincerely apologize for any lack of clarity in our original description and have now inserted the appropriate reference in the relevant section of the main text.

We deeply appreciate your valuable comments in identifying this ambiguity in our manuscript, which have significantly improved the accuracy and clarity of our work.

Line 283: what do you mean "may confer embryonic lethality"? Were they embryonic lethal or not?

We sincerely appreciate your critical request for clarification. Our experimental data from 15 pregnancies of *Zc3h11a* Het-KO mice intercrosses (n = 15 litters) conclusively confirmed the absence of homozygous knockout (Homo-KO) pups at birth. These findings align with the embryonic lethality of *Zc3h11a* homozygous deletion as reported by Younis et al. We fully acknowledge the ambiguity in our original phrasing and have revised the text to:“Second, *Zc3h11a* homozygous KO (Homo-KO) mice were not obtained in our study because homozygous deletion of exons confer embryonic lethality.”Your vigilance in ensuring terminological precision has greatly strengthened the rigor of our manuscript. We hope this clarification fully resolves your concerns.

Line 338: What is meant that Het-KO mice were constructed at 4 weeks of age? Do these mice not have a germline mutation?

Thank you for your valuable comments. We have revised the following content: “The germline heterozygous *Zc3h11a* knockout (Het-KO) mice were generated by CRISPR/Cas9-mediated gene editing at the embryonic stage on a C57BL/6J background, provided by GemPharmatech Co., Ltd (Nanjing, China). Phenotypic analyses were initiated when the mice reached four weeks of age.”

Line 346-347: how many mice were excluded due to having small eyes or ocular lesions? The methods section should state how refraction and ocular biometrics were measured.

Thank you for your valuable comments. We have added or revised the following content: “To exclude potential confounding effects of spontaneous ocular developmental abnormalities, a total of 7 mice (4 Het-KO and 3 WT) with small eyes or ocular lesions were excluded from the observation cohort. These anomalies were consistent with the baseline incidence of spontaneous malformations observed in historical colony data of wild-type C57BL/6J mice (approximately 11%), and were not attributed to the *Zc3h11a* heterozygous knockout.

The methods for measuring refraction and ocular biometrics are as follows and have been added to the original method. Refractive measurements were performed by a researcher blinded to the genotypes. Briefly, in a darkroom, mice were gently restrained by tail-holding on a platform facing an eccentric infrared retinoscope (EIR) (Schaeffel et al., 2004; Zhou et al., 2008a). The operator swiftly aligned the mouse position to obtain crisp Purkinje images centered on the pupil using detection software (Schaeffel et al., 2004), enabling axial measurements of refractive state and pupil size. Three repeated measurements per eye were averaged for analysis. The anterior chamber (AC) depth, lens thickness, vitreous chamber (VC) depth, and axial length (AL) of the eye were measured by real-time optical coherence tomography (a custom built OCT) (Zhou et al., 2008b). In simple terms, after anesthesia, each mouse was placed in a cylindrical holder on a positioning stage in front of the optical scanning probe. A video monitoring system was used to observe the eyes during the process. Additionally, by detecting the specular reflection on the corneal apex and the posterior lens apex in the two dimensional OCT image, the optical axis of the mouse eye was aligned with the axis of the probe. Eye dimensions were determined by moving the focal plane with a stepper motor and recording the distance between the interfaces of the eyes. Then, using the designed MATLAB software and appropriate refractive indices, the recorded optical path length was converted into geometric path length. Each eye was scanned three times, and the average value was taken.”

Line 428: what age retinas

Thank you for your meticulous attention to the experimental design details. Regarding the age of retinal samples, we have clarified the following in the revised manuscript:" Retinas were harvested from four-week-old mice for RNA sequencing." This revision enhances the transparency and reproducibility of our methodology. We deeply appreciate your rigorous review.

Figure 3 D-F: these images are too small to adequately assess, please show at higher magnification. Are there fewer bipolar cells, or just decreased expression of PKC? From these images, expression of ZC3H11A does not appear decreased, but the retina appears thinner. Is that true, or are these poorly matched sections?

Thank you for your professional insights regarding image quality and data interpretation. Your rigorous review has significantly enhanced the scientific rigor of our study. We hereby address your concerns point by point: The images in Figures 3D-F were acquired using a Zeiss LSM880 confocal microscope with a 10x eyepiece and 20x objective lens, a standard magnification for retinal section imaging that balances cellular resolution with full-thickness structural preservation. We quantified PKCα immunofluorescence intensity (a bipolar cell-specific marker) to assess changes in bipolar cell populations, rather than direct cell counting. This metric reflects PKCα expression abundance as a proxy for bipolar cell alterations (Figure 3H). To clarify terminology, we have revised the text to "bipolar cell-labelled protein PKCα immunofluorescence abundance" and detailed the methodology in the revised Methods section. Recognizing the semi-quantitative nature of fluorescence intensity analysis, we supplemented these data with Western blot results confirming reduced PKCα protein levels (Figure 3I). Zc3h11a expression was validated both by immunofluorescence intensity (Figure 3G) and Western blot (Figures 6F, H) quantification, confirming reduced expression in *Zc3h11a* Het-KO retinas. The apparent "retinal thinning" observed in histology sections stems from technical artifacts during tissue processing (fixation, dehydration, sectioning), not biological differences. HE staining, which better preserves sample morphology, showed no structural or thickness differences between Zc3h11a Het-KO mice and wild-type mice (Supplementary Figure 2).

Your expert feedback has driven us to establish a more robust validation framework. We believe the revised data now more accurately reflect the biological reality and sincerely hope these improvements meet your approval.

Figure 3G-J: Relative fluorescence intensity of immunohistochemistry is not a valid measure of protein expression.

We sincerely appreciate your thorough review and valuable comments regarding the immunofluorescence quantification method in Figures 3G-J. In response to your concern that "relative fluorescence intensity is not an effective quantitative measure of protein expression," we have implemented the following improvements to our analysis and validation: To ensure result reliability, all immunofluorescence experiments followed strict protocols: experimental and control samples were fixed, stained, and imaged in the same batch to eliminate inter-batch variability. Imaging was performed using a Zeiss LSM 880 confocal microscope with identical parameters, and the relative fluorescence intensity of specific signals per unit area was measured and statistically analyzed using ZEN software. We fully acknowledge the semi-quantitative nature of relative fluorescence intensity measurements. Therefore, we validated key differentially expressed proteins using Western blot analysis: The Western blot results for Zc3h11a (Figures 6F, H) were completely consistent with the relative fluorescence intensity trends (Figure 3G). Additionally, the newly included Western blot data for PKCα (Figure 3 I) further confirmed the reliability of our relative fluorescence intensity quantification. Your expert advice has significantly enhanced the rigor of our study. Should any additional data or clarification be required, we would be pleased to provide further support.

Figure 4: what are the arrows pointing at? This should be in the Figure legend. What is MB? Why are there no scale bars? What is difference between E and F, not clear from legend.

We sincerely appreciate your thorough review of Figure 4 and your valuable suggestions. In response to your concerns, we have carefully examined and improved the relevant content with the following modifications and clarifications: We sincerely apologize for not clearly indicating the arrow annotations in the original figure legend. In the revised version, we have provided detailed explanations for the arrow indicators: black arrows indicate perinuclear space dilation, blue arrows indicate cytoplasmic edema, and red arrows indicate disorganized and loosely arranged membrane discs. The updated legend has been clearly marked below Figure 4 in the main text. MB represents membrane discs, which are critical subcellular structures in the outer segments of retinal photoreceptor cells (rods and cones). They are responsible for light signal capture and transduction (containing visual pigments such as rhodopsin). The structural integrity of MB is essential for normal visual function. The scale bars in the original figures were located in the lower right corner of each subpanel, with specific parameters as follows: Figures 4A and B: magnification ×1000, scale bar 10 μm; Figures 4C and D: magnification ×700, scale bar 20 μm; Figures 4E and G: magnification ×2000, scale bar 5 μm; Figures 4F and H: magnification ×7000, scale bar 2 μm. Both Figures 4E and 4F show electron microscopy images of membrane discs (MB) in wild-type mouse photoreceptor cells. The only difference lies in the magnification: Figure 4E (×2000) demonstrates the overall arrangement pattern of membrane discs, while Figure 4F (×7000) focuses on ultrastructural details of the membrane discs (such as structural integrity). We have thoroughly checked the consistency between the figures and text, and have supplemented detailed legend descriptions in the main text. Once again, we sincerely appreciate your rigorous review, which has significantly enhanced the scientific rigor and readability of our study. Should you have any further suggestions, we would be happy to incorporate them.

Figure 5A: Why such a large y-axis? Figure legend does not match figure

We sincerely appreciate your careful review of Figure 5A and your valuable suggestions regarding the figure details. In response to your concerns, we have thoroughly examined and improved the relevant content as follows: The Y-axis of the volcano plot represents -log₁₀(p-value), where the magnitude of the values reflects statistical significance. Our RNA-seq data underwent rigorous multiple testing correction, and the adjusted p-values for some genes were extremely small, resulting in large values after -log₁₀ transformation. We have re-examined the data distribution and confirmed that the expanded Y-axis range is solely due to a small number of highly significant genes (as shown in the figure, the majority of genes remain clustered in the lower half of the Y-axis). This result accurately reflects the true data characteristics.

We sincerely apologize for the inadvertent error in the original labeling of "Up/Down" in the figure legend. This has now been corrected, and we strictly adhere to the following threshold criteria: Significantly upregulated (Up): adjusted p-value < 0.05 and log₂(FC) ≥ 1. Significantly downregulated (Down): adjusted p-value < 0.05 and log₂(FC) ≤ -1. To ensure the reliability of our conclusions, we have rechecked the raw data, statistical analysis, and visualization process. We confirmed that all significant genes strictly meet the above threshold criteria and that the visualization accurately reflects the true results. The revised figure has been updated in the manuscript as Figure 5A. We deeply appreciate your valuable feedback, which has helped us correct the errors in the figure and improve its accuracy and readability.

Figure 6F: Based on the western blot, only Zc3h11a appears different.

Thank you for your careful evaluation of the Western blot data in Figure 6F. We fully understand your concerns regarding the visual differences in PI3K and p-AKT/AKT bands and appreciate the opportunity to clarify the quantitative methodology and biological significance of these findings. Below we provide a detailed explanation of the experimental design and data analysis.

First, the data for each group were derived from retinal samples of three independent mice, with all experiments performed in parallel to control for technical variability. Image analysis was conducted using ImageJ software with standardized settings for grayscale quantification. Zc3h11a and PI3K levels were normalized to GAPDH as an internal reference, while p-AKT levels were calculated as a ratio to total AKT. The results showed that Zc3h11a protein levels were significantly reduced (p < 0.01, Figures 6F and H), consistent with the expected effects of heterozygous knockout, with good agreement between visual and statistical results. For PI3K and p-AKT/AKT, the bands appeared visually similar due to: The nonlinear nature of Western blot chemiluminescence signals in the saturation range, which compresses subtle quantitative differences in the images; the fact that p-AKT represents only 5-15% of the total AKT pool, making small proportional changes difficult to discern visually. However, it is important to note that both PI3K and p-AKT/AKT showed statistically significant differences between groups (p < 0.001 and p < 0.01, respectively; Figures 6G and I). Furthermore, signal transduction pathways exhibit cascade amplification effects - in the PI3K-AKT pathway, even small changes in upstream proteins can produce significant downstream effects (e.g., NF-κB activation) through kinase cascades (Figure 6J). Additionally, our RNA-Seq results revealed activation of the PI3K-AKT signaling pathway in *Zc3h11a* Het-KO mice (Figure 5D), and the qRT-PCR results were consistent with the western blot results (Figure 6A-C). Your expert comments have prompted us to present these data differences with greater biological rigor. Although the visual differences are subtle, based on statistical significance, pathway characteristics, and RNA sequencing, and qRT-PCR data, we believe these changes have biological relevance. We sincerely appreciate your commitment to data rigor and respectfully request your recognition of both the experimental results and the scientific logic of this study.

Figure 8: What is the role of ZC3H11A in this figure? Are the authors proposing that ZC3H11A regulates the translation of IκBα? They have not shown any evidence of that.

Thank you for your insightful exploration of the role of ZC3H11A in Figure 8. We appreciate your critical review and hope to elucidate the mechanistic framework behind our findings. In Figure 8, *Zc3h11a* is depicted as a regulator of IκBα mRNA nucleocytoplasmic transport, a mechanism originally elucidated by Darweesh et al. Their studies demonstrated that *Zc3h11a* binds to IκBα mRNA and promotes its nuclear export. Loss of *Zc3h11a* results in nuclear retention of IκBα mRNA, leading to reduced cytoplasmic IκBα protein levels and subsequent hyperactivation of the NF-κB pathway. While the specific nuclear export mechanism has been elucidated by Darweesh et al., our study demonstrates that *Zc3h11a* haploinsufficiency results in decreased IκBα mRNA and protein levels in the retina (Figure 7), linking *Zc3h11a* haploinsufficiency to NF-κB pathway dysregulation in myopia and highlighting that these molecular insights can be extended to a new pathological context (myopia). Your critical comments have enhanced the clarity of our mechanistic concepts and we hope that these descriptions will demonstrate the importance of *ZC3H11A* as a new candidate gene for myopia.